# Disruption in *CYLC1* leads to acrosome detachment, sperm head deformity, and male in/subfertility in humans and mice

Hui-Juan Jin[1†], Yong Fan[2†], Xiaoyu Yang[3†], Yue Dong[4], Xiao-Zhen Zhang[1], Xin-Yan Geng[1], Zheng Yan[2], Ling Wu[2], Meng Ma[2], Bin Li[2], Qifeng Lyu[2], Yun Pan[4], Mingxi Liu[5]*, Yanping Kuang[2]*, Su-Ren Chen[1]*

[1]Key Laboratory of Cell Proliferation and Regulation Biology, Ministry of Education, Department of Biology, College of Life Sciences, Beijing Normal University, Beijing, China; [2]Department of Assisted Reproduction, Shanghai Ninth People's Hospital, Shanghai Jiao Tong University School of Medicine, Shanghai, China; [3]State Key Laboratory of Reproductive Medicine and Offspring Health, The Center for Clinical Reproductive Medicine, The First Affiliated Hospital of Nanjing Medical University, Nanjing, China; [4]State Key Laboratory of Reproductive Medicine and Offspring Health, Department of Histology and Embryology, School of Basic Medical Sciences, Nanjing Medical University, Nanjing, China; [5]State Key Laboratory of Reproductive Medicine and Offspring Health, The Affiliated Taizhou People's Hospital of Nanjing Medical University, Taizhou School of Clinical Medicine, Nanjing Medical University, Nanjing, China

*For correspondence:
mingxi.liu@njmu.edu.cn (ML);
kuangyanp@126.com (YK);
chensr@bnu.edu.cn (SRC)

†These authors contributed equally to this work

**Abstract** The perinuclear theca (PT) is a dense cytoplasmic web encapsulating the sperm nucleus. The physiological roles of PT in sperm biology and the clinical relevance of variants of PT proteins to male infertility are still largely unknown. We reveal that cylicin-1, a major constituent of the PT, is vital for male fertility in both mice and humans. Loss of cylicin-1 in mice leads to a high incidence of malformed sperm heads with acrosome detachment from the nucleus. Cylicin-1 interacts with itself, several other PT proteins, the inner acrosomal membrane (IAM) protein SPACA1, and the nuclear envelope (NE) protein FAM209 to form an 'IAM–cylicins–NE' sandwich structure, anchoring the acrosome to the nucleus. WES (whole exome sequencing) of more than 500 Chinese infertile men with sperm head deformities was performed and a *CYLC1* variant was identified in 19 patients. *Cylc1*-mutant mice carrying this variant also exhibited sperm acrosome/head deformities and reduced fertility, indicating that this *CYLC1* variant most likely affects human male reproduction. Furthermore, the outcomes of assisted reproduction were reported for patients harbouring the *CYLC1* variant. Our findings demonstrate a critical role of cylicin-1 in the sperm acrosome–nucleus connection and suggest *CYLC1* variants as potential risk factors for human male fertility.

## eLife assessment

Spermiogenesis is a complex process allowing the emergence of specific sperm organelle, including the acrosome, a sperm giant vesicle of secretion. This **important** study reports the key role of Cylicin-1 in acrosome biogenesis and identifies the molecular partners necessary for acrosome anchoring. The **compelling** demonstration is based on infertile patient samples and two animal models. Overall, this provides results that will be invaluable to the male reproduction community, including scientists and andrologists.

## Introduction

In mammalian sperm, the cytoskeletal element encapsulating the nuclei is a detergent- and high salt-resistant structure called the perinuclear theca (PT) (*Oko and Sutovsky, 2009*). Apically, the PT resides between the inner acrosomal membrane (IAM) and the nuclear envelope (NE) making up the subacrosomal layer (PT-SAL, also known as the acroplaxome), while caudally, it resides between the cell membrane and the NE making up the postacrosomal part (PT-PAR, also known as the calyx). The PT is not composed of traditional cytoskeletal proteins but rather of a variety of spermatid-specific cytosolic and nuclear proteins. Longo et al. determined that two kinds of basic proteins are the major constituents of PT: calicin and cylicins (*Longo et al., 1987*). After that, more than 20 PT-enriched proteins, such as fatty acid-binding protein (FABP)9 (*Oko and Morales, 1994*), actin-capping protein (CP) α3/β3 (*von Bülow et al., 1997*), transcription factor STAT4 (*Herrada and Wolgemuth, 1997*), actin-related proteins ACTRT1/2 (*Heid et al., 2002*), histone H2B variant H2BL1 (*Aul and Oko, 2002*), sperm-borne oocyte activating factors (PLC ζ and PAWP) (*Aarabi et al., 2014*; *Hachem et al., 2017*), and the glutathione *S*-transferase GSTO2 (*Hamilton et al., 2017*), have been identified from mammalian sperm.

The PT is speculated to function as (1) an architectural element involved in the shape changes of the nucleus during spermiogenesis, (2) a cement linking the acrosome with the nucleus, and (3) a candidate for sperm-borne oocyte activation. However, the physiological roles of PT in sperm are largely uncertain due to the lack of evidence from gene knockout animal models. Our recent studies of *Ccin*- and *Actrt1*-knockout mice indicate that PT, at least, physiologically functions to protect nuclear structure and anchor developing acrosomes to the nucleus (*Zhang et al., 2022a*; *Zhang et al., 2022b*). Loss of calicin specifically leads to the surface subsidence of sperm heads in the nuclear condensation stage (*Zhang et al., 2022a*). Importantly, we further report that three infertile men with sperm head deformities carry homozygous pathogenic mutations in *CCIN*, and the corresponding *Ccin*-mutant mice mimic the phenotype in patients (*Fan et al., 2022*). *Actrt1*-KO male mice are severely subfertile as a result of detached acrosomes and fertilization deficiency (*Zhang et al., 2022b*). ACTRT1 anchors acrosomes to the nucleus by interacting with some IAM proteins and NE proteins (*Zhang et al., 2022b*).

Hess et al. cloned cDNAs encoding cylicin-1, named for the Greek word for 'cup' or 'beaker'. 74 kDa cylicin-1 contains numerous lysine dipeptides followed by a third variable amino acid (KKX). The C-terminal tail contains proline-rich segments and the central portion of the protein is arranged as a series of repeating units that are predicted to form short alpha helices. Northern blotting detects the cylicin-1 transcript only in the testis, and an antibody against cylicin-1 specifically stains the PT-PAR of human and bovine sperm (*Hess et al., 1993*). Apart from the first expression study of calicin-1 published 30 years ago, the physiological role of cylicin-1 in sperm biology is still unknown.

In this study, we generated a *Cylc1*-KO mouse model and revealed that loss of cylicin-1 leads to severe male subfertility as a result of sperm head deformities and acrosome detachment. A *CYLC1* variant was identified from a WES (whole exome sequencing) study of infertile patients with sperm head deformities. The contribution of this variant to male subfertility with sperm acrosome detachment is further confirmed by animal evidence from *Cylc1*-mutant mice.

## Results

### Generation of *Cylc1*-KO mice

Cylicin-1 is a major and specific constituent of the PT of mammalian sperm (*Longo et al., 1987*). According to the Expression Atlas database, cylicin-1 was specifically expressed in testis tissues, and restricted to early and late spermatid subpopulations (*Figure 1A* and *Figure 1—figure supplement 1*). Using our homemade rabbit anti-CYLC1 antibody (*Figure 1—figure supplement 2*), we showed that cylicin-1 was initially expressed at the subacrosomal layer (SAL) in round and elongating spermatids and then translocated to the postacrosomal region (PAR) in sperm (*Figure 1B*). The dynamic expression pattern of cylicin-1 is quite similar to that of CAPZA3 (*von Bülow et al., 1997*), calicin (*Zhang et al., 2022a*), and ACTRT1 (*Zhang et al., 2022b*), all showing SAL-to-PAR translocation during spermiogenesis.

To explore the physiological role of cylicin-1, we disrupted its function in mice by applying CRISPR/Cas9 technology (*Figure 1C*). The *Cylc1* gene has only one transcript and a 3651-bp region containing

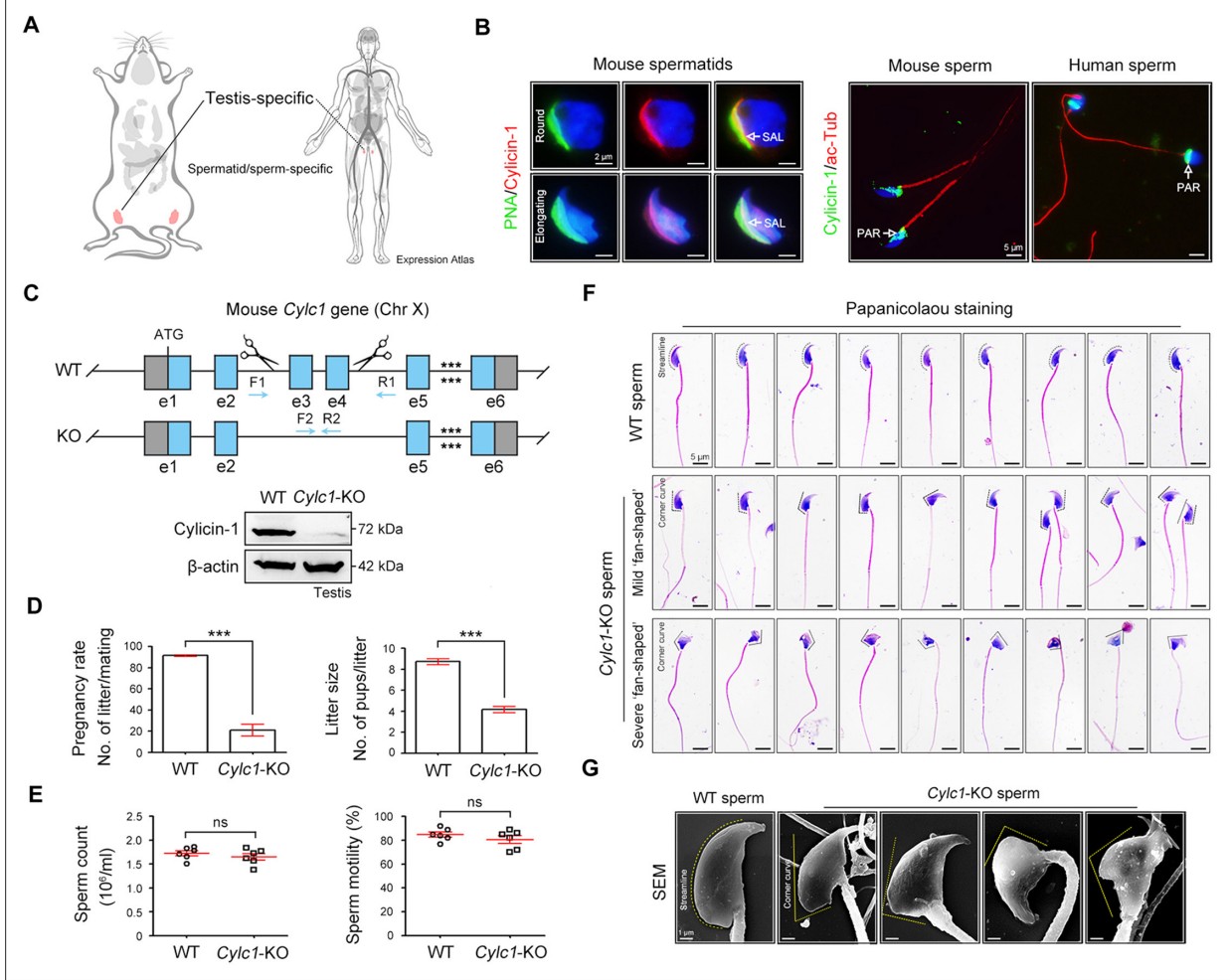

**Figure 1.** Severe male subfertility with sperm head deformity in *Cylc1*-KO mice. (**A**) According to the Expression Atlas database (https://www.ebi.ac.uk/gxa/home), cylicin-1 was specifically expressed in the testis. Within the testis, cylicin-1 was restricted to the population of early and late spermatids. (**B**) Costaining of cylicin-1 and peanut agglutinin (PNA)-FITC (fluorescein Isothiocyanate) (an acrosome dye) in mouse round and elongating spermatids (left). Scale bars, 2 μm. Costaining of cylicin-1 and acetylated-tubulin (a flagella marker) in mouse and human sperm (right). Scale bars, 5 μm. Nuclei were counterstained with DAPI (4',6-diamidino-2-phenylindole). SAL, subacrosomal layer; PAR, postacrosomal region. (**C**) Schematic illustration of the targeting strategy used to generate *Cylc1*-KO mice. DNA extracted from mouse tails was used for PCR (polymerase chain reaction) genotyping. Knockout efficiency was determined by western blotting using testis protein lysates. β-Actin served as a loading control. (**D**) Adult *Cylc1*-KO male mice and their littermate wild-type (WT) mice (*n* = 3 each group) were continuously coupled with WT female mice at a ratio of 1:2 for 2 months. Data are represented as the mean ± standard error of the mean (SEM), Student's *t* test, ***p < 0.001. (**E**) Sperm counts were determined using a fertility counting chamber under a light microscope, and total motility was assessed via computer-assisted semen analysis (CASA). Data are represented as the mean ± SEM (*n* = 6 each group), Student's *t* test, ns, not significant. (**F**) Light microscopy analysis of sperm from WT mice and *Cylc1*-KO mice (*n* = 3 each) with Papanicolaou staining. The dashed curve indicates the streamline surface at the dorsal side of sperm heads of WT mice. The dashed broken lines point to the corner curve at the dorsal side of sperm heads of *Cylc1*-KO mice. Scale bars, 5 μm. (**G**) Scanning electron microscopy (SEM) images of sperm from WT mice and *Cylc1*-KO mice. Scale bars, 1 μm.

The online version of this article includes the following source data and figure supplement(s) for figure 1:

**Source data 1.** Original file for the western blot in *Figure 1C*.

**Source data 2.** PDF containing *Figure 1C* and relevant western blot with highlighted bands and sample labels.

**Source data 3.** Data used for analysis of *Figure 1D*.

**Source data 4.** Data used for analysis of *Figure 1E*.

**Figure supplement 1.** Expression pattern of human *CYLC1* mRNA among different tissues and within the testis.

**Figure supplement 2.** Generation of cylicin-1 antiserum.

**Figure supplement 2—source data 1.** Original file for the western blot in *Figure 1—figure supplement 2*.

*Figure 1 continued on next page*

*Figure 1 continued*

**Figure supplement 2—source data 2.** PDF containing *Figure 1—figure supplement 2* and relevant western blot with highlighted bands and sample labels.

**Figure supplement 3.** Construction report of *Cylc1*-KO mice.

**Figure supplement 4.** Analysis of the reproductive system of *Cylc1*-KO male mice.

**Figure supplement 4—source data 1.** Data used for analysis of *Figure 1—figure supplement 4*.

exons 3 and 4 was selected as the knockout region (*Figure 1—figure supplement 3*). The *Cylc1* gene is located on the X-chromosome; accordingly, female *Cylc1*$^{+/-}$ founder mice were crossed with wild-type (WT) males to generate *Cylc1*$^{-/Y}$ (hereafter called *Cylc1*-KO) mice, which were tested by genotyping PCR of tail genomic DNA (*Figure 1C*). Western blotting analysis with anti-cylicin-1 antibody confirmed absence of the cylicin-1 in the testis protein lysates of *Cylc1*-KO mice (*Figure 1C*).

## Severe male subfertility of *Cylc1*-KO mice

Adult *Cylc1*-KO male mice were normal in appearance, healthy and developed with no identifiable abnormalities. Given that cylicin-1 is a sperm-specific protein, we studied the fertility of *Cylc1*-KO male mice by mating them with WT females. Although *Cylc1*-KO male mice were able to mate normally with females, they suffered from severe male subfertility. Both the pregnancy rates of mating females and pups that were born from *Cylc1*-KO males were significantly reduced compared with those from WT males (*Figure 1D*). The testis size and weight were similar between *Cylc1*-KO mice and their littermate WT mice (*Figure 1—figure supplement 4*). Histological examination of *Cylc1*-KO testis and cauda epididymis revealed generally normal spermatogenesis and sperm production (*Figure 1—figure supplement 4*). Both the sperm count and the total motility were similar between *Cylc1*-KO mice and WT mice (*Figure 1E*).

We next examined whether Cylc1-KO mice suffer from sperm morphological abnormalities. Sperm were obtained from the cauda epididymis and analysed by Papanicolaou staining. The sperm of WT mice showed a typical 'hook-shaped' appearance with a streamlined dorsal part (*Figure 1F*), whereas the Cylc1-KO mice produced sperm with abnormal head morphologies, exhibiting varying degrees of 'fan-shaped' heads (*Figure 1F*). Mild 'fan-shaped' heads showed a roughly similar morphology to normal sperm heads but with an obvious corner curve at the dorsal side (*Figure 1F*, second row), while severe 'fan-shaped' heads exhibited a quite different morphology and looked like a 'fan' when observed from the head–tail connection (*Figure 1F*, third row). Sperm with mild and severe 'fan-shaped' heads accounted for approximately 60% and 40% of the total sperm in Cylc1-KO mice, respectively. The phenotype of 'fan-shaped' heads in sperm of Cylc1-KO mice was further identified by scanning electron microscopy (*Figure 1G*).

## Acrosome detachment in spermatids and sperm of Cylc1-KO mice

To explore the reason for the sperm head deformities, we examined the acrosome structure in *Cylc1*-KO mice by peanut agglutinin (PNA)-FITC staining. In spermatids of WT mice, the developing acrosome was tightly bound to the NE of the sperm nucleus. Spermatids of *Cylc1*-KO mice, by contrast, exhibited varying degrees of acrosome detachment from the NE (*Figure 2A*). Statistical analysis indicated that the ratio of acrosome detachment in spermatids of *Cylc1*-KO mice was significantly higher than that in WT mice (*Figure 2B*). We further examined PT-SAL, an F-actin-containing plate between the IAM and NE, using an F-actin fluorescence probe. Compared with WT mice, spermatids of *Cylc1*-KO mice showed a disturbance of the PT-SAL distribution, and sometimes it broke away from the NE (*Figure 2C*). The percentage of spermatids with abnormal PT-SAL structure in *Cylc1*-KO mice was also significantly greater than that in WT mice (*Figure 2D*). Under a transmission electron microscope (TEM), the phenomenon of acrosome detachment in the sperm of *Cylc1*-KO mice was clearly observed (*Figure 2E*). The number of sperm with detached acrosomes was significantly higher in *Cylc1*-KO mice than in WT mice (*Figure 2F*).

To explore whether acrosome detachment occurs during spermiogenesis, we performed TEM analysis of elongating spermatids within the testis. TEM analysis clearly revealed an enlarged space area between the IAM and the NE in spermatids of *Cylc1*-KO mice (*Figure 2G*). The average PT-SAL thickness (space between the IAM and the NE) was significantly larger in the spermatids of *Cylc1*-KO

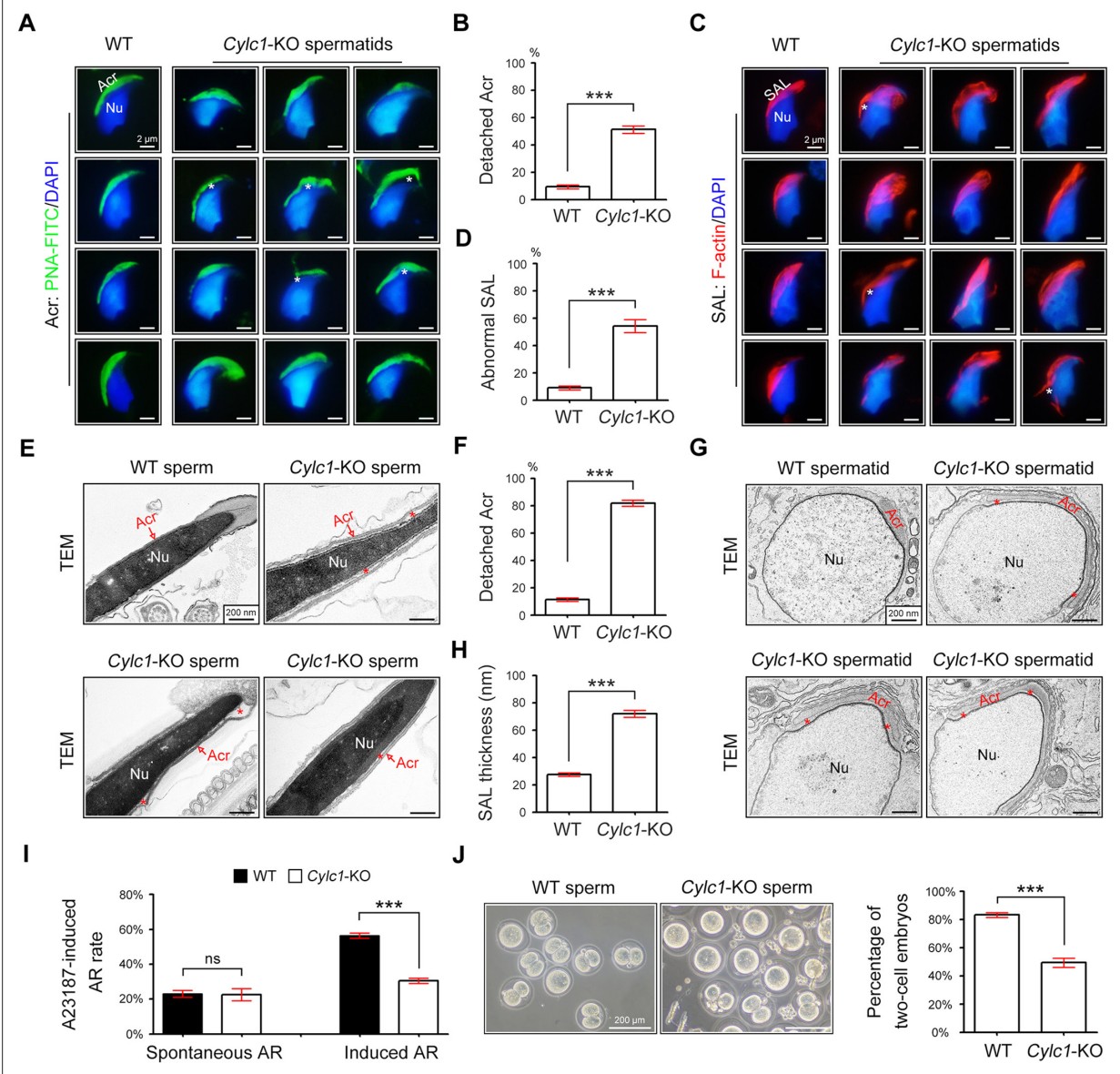

**Figure 2.** Acrosome detachment in the spermatids and sperm of *Cylc1*-KO mice. (**A**) Visualization of the acrosome (Acr) in spermatids of wild-type (WT) and *Cylc1*-KO mice using the fluorescent dye peanut agglutinin (PNA)-FITC. The nuclei (Nu) were counterstained with DAPI dye. Asterisks indicate detached acrosomes. Scale bars, 2 µm. (**B**) Percentage of spermatids with detached Acr in WT and *Cylc1*-KO mice. (**C**) Subacrosomal layer of perinuclear theca (PT-SAL) actin bundles in spermatids of WT and *Cylc1*-KO mice was visible by an F-actin-Tracker Red dye. The asterisk indicates the detachment of Apx from Nu. Scale bars, 2 µm. (**D**) Percentage of spermatids with abnormal Apx structure in WT and *Cylc1*-KO mice. In B and D, 100 spermatids in each group (n = 3) were counted. Data are represented as the mean ± standard error of the mean (SEM), Student's *t* test, ***p < 0.001. (**E**) Transmission electron microscopy (TEM) analysis showing detachment of Acr from Nu in the sperm of *Cylc1*-KO mice. The asterisk indicates the enlarged space area between the Acr and Nu. Scale bars, 200 nm. (**F**) Percentage of sperm with detached Acr in WT and *Cylc1*-KO mice as revealed by TEM. Twenty-five sperm in each group (n = 3) were counted. (**G**) TEM analysis of testis tissues revealing detachment of the developing Acr from Nu in spermatids of *Cylc1*-KO mice. The asterisk indicates the enlarged space area between the Acr and Nu. Scale bars, 200 nm. (**H**) Apx thickness in the sperm of WT mice and *Cylc1*-KO mice. The thickness of the Apx was measured by distance meter software of a Tecnai G2 Spirit electron microscope in TEM images at a magnification of ×30,000. Five spermatids were measured in each group (n = 3). Data in F and H are represented as the mean ± SEM, Student's *t* test, ***p < 0.001. (**I**) Spontaneously underwent and A23187-induced acrosome reaction (AR). PNA-FITC dye was used to label acrosomes, and the sperm nuclei were stained with DAPI. One hundred sperm in each group (n = 3) were counted. Data are represented as the mean ± SEM, Student's *t* test, ns, not significant; ***p < 0.001. (**J**) An in vitro fertilization assay was performed using sperm from *Cylc1*-KO mice and WT mice (n = 3 for each group). The percentage of two-cell embryos was calculated at 24 hr postfertilization. Data are represented as the mean ± SEM, Student's *t* test, ***p < 0.001.

The online version of this article includes the following source data and figure supplement(s) for figure 2:

**Source data 1.** Data used for analysis of *Figure 2B*.

*Figure 2 continued on next page*

*Figure 2 continued*

**Source data 2.** Data used for analysis of *Figure 2D*.

**Source data 3.** Data used for analysis of *Figure 2F*.

**Source data 4.** Data used for analysis of *Figure 2H*.

**Source data 5.** Data used for analysis of *Figure 2I*.

**Source data 6.** Data used for analysis of *Figure 2J*.

**Figure supplement 1.** Normal development of manchette in spermatids of *Cylc1*-KO mice.

mice than those of in WT mice (*Figure 2H*). Moreover, the manchette was generally normal in the spermatids of *Cylc1*-KO mice, as revealed by tubulin fluorescence probe staining and TEM analysis (*Figure 2—figure supplement 1*).

We next determined whether sperm acrosome detachment in *Cylc1*-KO mice affects the acrosome reaction (AR) that is required for physiological fertilization. Intact acrosomes can be labelled with PNA-FITC dye, whereas acrosome-reacted sperm lack PNA-FITC signals (*Hao et al., 2014*). There was no difference in the ratio of spontaneous AR in sperm between WT mice and *Cylc1*-KO mice; however, the calcium ionophore A23187-induced AR rate in sperm was significantly lower in *Cylc1*-KO mice than in WT mice (*Figure 2I*). We next performed in vitro fertilization (IVF) using sperm from *Cylc1*-KO mice or WT mice. The percentage of two-cell embryos using sperm of *Cylc1*-KO mice was significantly decreased compared with that of WT mice (*Figure 2J*).

## Cylicin-1 interacts with itself and other PT proteins

To explore the molecular mechanisms underlying the acrosome–nucleus connection by calicin-1, the interacting partners of calicin-1 were studied using protein–protein interaction technologies. First, we found that FLAG-tagged CYLC1 could be coimmunoprecipitated with Myc-tagged CYLC1 in HEK293T cells, demonstrating that cylicin-1 interacts with itself (*Figure 3—figure supplement 1*). The interactions between Myc-tagged CYLC1 and several other FLAG-tagged PT proteins, such as ACTRT1, ACTRT2, ACTL7A, CAPZA3, and CCIN, were further revealed by coimmunoprecipitation (co-IP) assays in HEK293T cells (*Figure 3A* and *Figure 3—figure supplement 1*). The ACTL7A–CYLC1 interaction was further confirmed by endogenous co-IP assay using mouse testis extracts (*Figure 3B*). Moreover, microscale thermophoresis (MST) analysis indicated a binding affinity between GFP-labelled CYLC1 and purified ACTL7A in a dose-dependent manner (*Figure 3C*). The assembly of these PT proteins into a large protein complex suggests that they may play a coordinated role in acrosome attachment. The protein levels of PT proteins, such as ACTRT1, ACTRT2, ACTL7A, CCIN, CAPZA3, CAPZB, CYLC2, and PLC ζ , were similar in the sperm sample between WT mice and *Cylc1*-KO mice, indicating that the sperm acrosome detachment in *Cylc1*-KO mice may not be due to altered PT protein contents (*Figure 3A* and *Figure 3—figure supplement 2*). Previous studies have indicated that ACTRT1 interacts with ACTL7A to form sperm PT-specific actin-related proteins (ARPs), and their knockout/mutant mice show acrosome detachment (*Xin et al., 2020*; *Fan et al., 2022*). We found that the endogenous interaction between ACTRT1 and ACTL7A was partially disrupted by the loss of cylicin-1 (*Figure 3D*), highlighting the assistant role of cylicin-1 on the ACTRT1–ACTL7A connection.

## Cylicin-1 connects with NE protein FAM209 and IAM protein SPACA1

Given that PT-SAL is physically located between the NE and the IAM, cylicin-1 proteins may interact with NE and IAM proteins to anchor the developing acrosome to the nucleus. We showed that Myc-tagged CYLC1 was immunoprecipitated with the FLAG-tagged NE protein FAM209 in HEK293T cells (*Figure 3E*). The interaction between CYLC1 and FAM209 was further confirmed by endogenous co-IP using mouse testis extracts (*Figure 3F*) and MST analysis using GFP-FAM209 cell lysates and purified CYLC1 (*Figure 3G*). FAM209 is an interacting partner of DPY19L2, and loss of the PT-SAL association and abnormal acrosome development are observed in its knockout mice (*Castaneda et al., 2021*). We further found that FLAG-tagged CYLC1 was immunoprecipitated with the Myc-tagged IAM protein SPACA1 in HEK293T cells (*Figure 3H*). The interaction between CYLC1 and SPACA1 was further confirmed by endogenous co-IP using mouse testis extracts (*Figure 3I*) and a dose-dependent binding affinity between GFP-SPACA1 from cell lysates and recombinant CYLC1 by MST (*Figure 3J*). *Spaca1*-deficient spermatids exhibit an attenuated association of the IAM with the NE (*Fujihara et al.,*

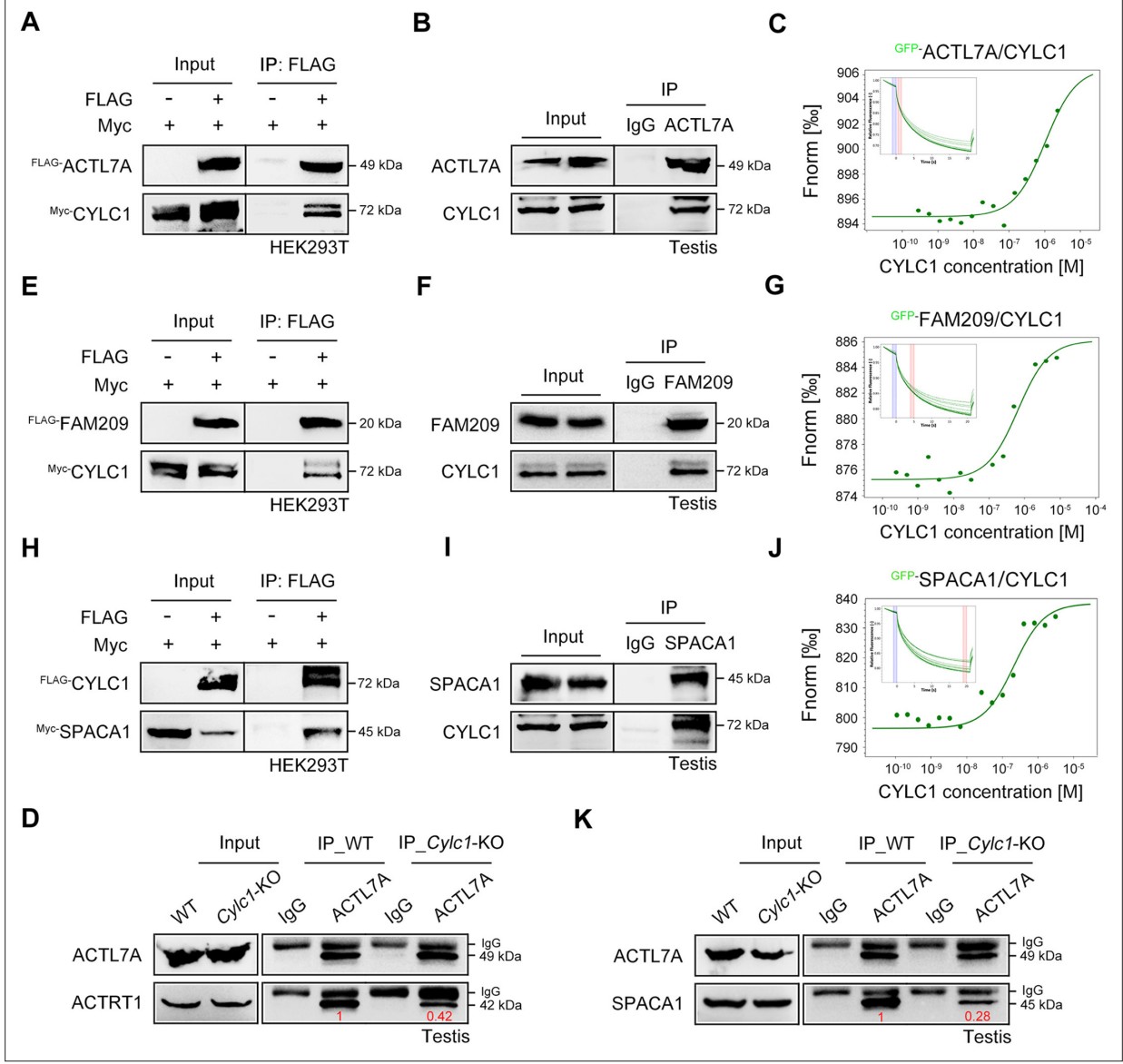

**Figure 3.** Roles of cylicin-1 in the formation of nuclear envelope (NE)–subacrosomal layer (SAL)–inner acrosomal membrane (IAM) protein complexes. (**A**) Myc-tagged CYLC1 was transfected into HEK293T cells with or without FLAG-tagged ACTL7A. Protein lysates were immunoprecipitated with anti-FLAG antibody and then detected with anti-Myc antibody by sodium dodecyl sulphate–polyacrylamide gel electrophoresis (SDS–PAGE). (**B**) Mouse testis lysates were immunoprecipitated with anti-ACTL7A antibody or rabbit IgG and then detected with anti-CYLC1 antibody. (**C**) The binding affinity between different concentrations of purified CYLC1 and GFP-ACTL7A cell lysates was measured by microscale thermophoresis (MST). (**D**) Testis protein lysates from wild-type (WT) and *Cylc1*-KO mice were immunoprecipitated with anti-ACTL7A antibody and then detected with anti-ACTRT1 antibody. (**E**) Myc-tagged CYLC1 was immunoprecipitated with FLAG-tagged FAM209 in HEK293T cell extracts. (**F**) Mouse testis protein lysates were immunoprecipitated with anti-FAM209 antibody or rabbit IgG and then detected with anti-CYLC1 antibody. (**G**) The binding affinity between purified CYLC1 and GFP-FAM209 cell lysates was revealed by an MST assay. (**H**) Myc-tagged SPACA1 was immunoprecipitated with FLAG-tagged CYLC1 in HEK293T cell extracts. (**I**) Mouse testis protein lysates were immunoprecipitated with anti-SPACA1 antibody or rabbit IgG and then detected with anti-CYLC1 antibody. (**J**) Binding between GFP-SPACA1 from HEK293T cell lysates and recombinant CYLC1. (**K**) Testis protein lysates from WT mice or *Cylc1*-KO mice were incubated with anti-ACTL7A antibody and then detected with anti-SPACA1 antibody.

The online version of this article includes the following source data and figure supplement(s) for figure 3:

**Source data 1.** Original file for the western blot in *Figure 3A*.

**Source data 2.** PDF containing *Figure 3A* and relevant western blot with highlighted bands and sample labels.

**Source data 3.** Original file for the western blot in *Figure 3B*.

**Source data 4.** PDF containing *Figure 3B* and relevant western blot with highlighted bands and sample labels.

*Figure 3 continued*

**Source data 5.** Original file for the western blot in *Figure 3D*.

**Source data 6.** PDF containing *Figure 3D* and relevant western blot with highlighted bands and sample labels.

**Source data 7.** Original file for the western blot in *Figure 3E*.

**Source data 8.** PDF containing *Figure 3E* and relevant western blot with highlighted bands and sample labels.

**Source data 9.** Original file for the western blot in *Figure 3F*.

**Source data 10.** PDF containing *Figure 3F* and relevant western blot with highlighted bands and sample labels.

**Source data 11.** Original file for the western blot in *Figure 3H*.

**Source data 12.** PDF containing *Figure 3H* and relevant western blot with highlighted bands and sample labels.

**Source data 13.** Original file for the western blot in *Figure 3I*.

**Source data 14.** PDF containing *Figure 3I* and relevant western blot with highlighted bands and sample labels.

**Source data 15.** Original file for the western blot in *Figure 3K*.

**Source data 16.** PDF containing *Figure 3K* and relevant western blot with highlighted bands and sample labels.

**Figure supplement 1.** CYLC1 interacts with itself and several other perinuclear theca (PT) proteins.

**Figure supplement 1—source data 1.** Original file for the western blot in *Figure 3—figure supplement 1A*.

**Figure supplement 1—source data 2.** PDF containing *Figure 3—figure supplement 1A* and relevant western blot with highlighted bands and sample labels.

**Figure supplement 1—source data 3.** Original file for the western blot in *Figure 3—figure supplement 1B*.

**Figure supplement 1—source data 4.** PDF containing *Figure 3—figure supplement 1B* and relevant western blot with highlighted bands and sample labels.

**Figure supplement 1—source data 5.** Original file for the western blot in *Figure 3—figure supplement 1C*.

**Figure supplement 1—source data 6.** PDF containing *Figure 3—figure supplement 1C* and relevant western blot with highlighted bands and sample labels.

**Figure supplement 1—source data 7.** Original file for the western blot in *Figure 3—figure supplement 1D*.

**Figure supplement 1—source data 8.** PDF containing *Figure 3—figure supplement 1D* and relevant western blot with highlighted bands and sample labels.

**Figure supplement 1—source data 9.** Original file for the western blot in *Figure 3—figure supplement 1E*.

**Figure supplement 1—source data 10.** PDF containing *Figure 3—figure supplement 1E* and relevant western blot with highlighted bands and sample labels.

**Figure supplement 2.** Expression of perinuclear theca (PT) proteins in sperm samples of wild-type (WT) mice and *Cylc1*-KO mice.

**Figure supplement 2—source data 1.** Original file for the western blot in *Figure 3—figure supplement 2A*.

**Figure supplement 2—source data 2.** PDF containing *Figure 3—figure supplement 2A* and relevant western blot with highlighted bands and sample labels.

**Figure supplement 2—source data 3.** Data used for analysis of *Figure 3—figure supplement 2B*.

*2012*). SPACA1 has also been shown to interact with ACTL7A to anchor the developing acrosome to the PT-SAL (*Chen et al., 2021*). Loss of cylicin-1 partially disrupted the IAM and PT-SAL connections because the interaction between SPACA1 and ACTL7A was weakened in *Cylc1*-KO mice, as revealed by an endogenous co-IP assay using mouse testis extracts (*Figure 3K*).

## Identification of *CYLC1* variants in infertile patients with sperm head deformities

To explore the clinical relevance of cylicin-1 deficiency to human male infertility, we performed WES analysis in a large cohort of more than 500 infertile patients with sperm head deformities (*Figure 4A*). These patients were recruited from the Shanghai Ninth People's Hospital of Shanghai Jiao Tong University and the First Affiliated Hospital of Nanjing Medical University. Intron mutations, synonymous mutations, variants with allele frequency >1% in the Genome Aggregation Database (gnomAD, total population) (*Gudmundsson et al., 2022*), and 'benign' or 'likely benign' variants according to the American College of Medical Genetics and Genomics (ACMG) (*Gonzales et al., 2022*), were excluded. Testis-specifically or testis-predominantly expressed genes were retained. We identified some patients carrying homozygous or compound heterozygous mutations in six known sperm head

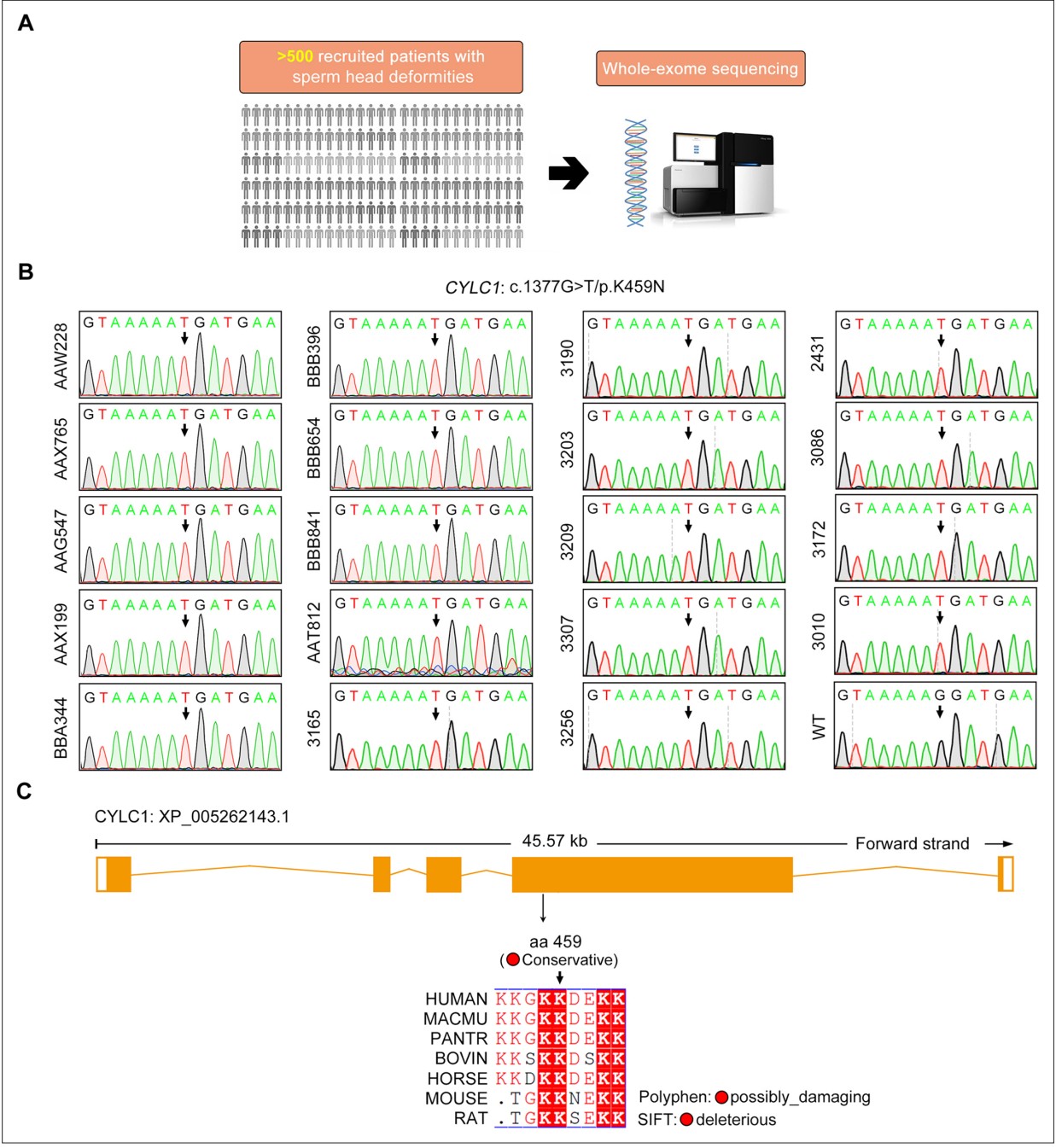

**Figure 4.** Identification of a *CYLC1* variant in a large cohort of Chinese infertile men with sperm head deformities. (**A**) More than 500 patients with sperm head deformities were recruited for WES and subsequent bioinformatic analysis. Nineteen individuals harboured a variant of CYLC1. (**B**) Sanger sequencing of the 19 individuals who were affected by the c.1377G>T/p.K459N variant in the *CYLC1* gene. Sanger sequencing of the wild-type (WT) allele in healthy controls is also shown. (**C**) This variant was located at the fourth exon of the *CYLC1* gene. Conservation analysis was performed by Basic Local Alignment Search Tool (BLAST). The NCBI protein ID for CYLC1: NP_066941.1 (Human), XP_001102833.1 (*Macaca mulatta*), XP_016799824.1 (*Pan troglodytes*), NP_776727.1 (Bovin), XP_023489864.1 (Horse), NP_080410.3 (Mouse), and XP_006229368.1 (Rat). The pathogenicity of variants was analysed by Polyphen2 and SIFT.

deformity-associated genes, including *DPY19L2* (*Koscinski et al., 2011*), *ZPBP1* (*Yatsenko et al., 2012*), *PICK1* (*Liu et al., 2010*), *ACTL7A* (*Xin et al., 2020*), *ACTL9* (*Dai et al., 2021*), and *CCIN* (*Zhang et al., 2022a*; *Supplementary file 1*). Intriguingly, 19 individuals were identified to harbor a variant of CYLC1 (GenBank: NM_021118.3): c.1377G>T/p.K459N (*Figure 4B*). The variant was predicted to be deleterious through utilization of SIFT (*Ng and Henikoff, 2003*), PolyPhen-2 (*Adzhubei et al., 2013*),

**Table 1.** Nineteen individuals carrying a *CYLC1* variant identified from >500 infertile men with sperm head deformities.

| | Variant |
|---|---|
| cDNA alternation | c.1377G>T |
| Protein alteration | p.K459N |
| Patient(s) | AAG547; BBB654; BBB841; AAX199; AAX765; BBA344; BBB396; AAW228; AAT812; 3086; 3165; 3172; 3190; 3203; 3209; 3307; 3256; 3010; 2431 |
| | **Allele frequency in human population** |
| gnomAD genome | 0.001386 |
| | **Functional prediction** |
| SIFT | Deleterious |
| PolyPhen-2 | Damaging |
| CADD | 15.18 |
| | **Conservation between mouse and human** |
| | Yes |

NCBI reference sequence number of CYLC1 is GenBank: NM_021118.3.

and CADD (*Rentzsch et al., 2019*; *Table 1*). Moreover, the amino acid residue of this variant was conserved between humans and mice (*Figure 4C*).

## Sperm acrosome detachment in men harbouring the CYLC1 variant

The rates of morphological normality of sperm decreased to zero in 7 of 19 patients harbouring CYLC1 variant, and the remaining affected individuals had a morphological normality rate of less than 4% (*Barratt et al., 2017*; *Table 2*). The morphology of sperm heads and acrosomes in men harboring the CYLC1 variant was assessed with Papanicolaou staining (*Figure 5A*), immunofluorescence costaining (*Figure 5B*), and TEM analysis (*Figure 5C*). In the healthy control, the acrosome covered more than one-third of the anterior sperm head; in contrast, sperm from men harbouring CYLC1 variant displayed deformed acrosomes (*Figure 5A*). Costaining of PNA-FITC and cylicin-1 indicated that the PT-PAR distribution of cylicin-1 was disrupted and the acrosomes were deformed in sperm from patients harbouring CYLC1 variant (*Figure 5B*). TEM analysis clearly revealed that acrosomes were detached in sperm from patients harbouring CYLC1 variant (*Figure 5C*).

## *Cylc1*-mutant mice exhibit male subfertility with detached acrosomes of sperm

We next generated *Cylc1*-mutant mice that carried a single amino acid change equivalent to the variant in human CYLC1 using CRISPR/Cas9 technology (*Figure 6A, B*). *Cylc1*-mutant mice grew normally but were male subfertile. Of the 42 female mice mated with WT males, 35 were pregnant and gave rise to 281 offspring. In contrast, of the 28 female mice mated with *Cylc1*-mutant males, only 11 were pregnant, giving rise to 49 offspring (*Figure 6C, D*). *Cylc1*-mutant mice and WT mice showed similar testis sizes and weights (*Figure 6—figure supplement 1*). Spermatogenesis and sperm production were generally normal, as indicated by hematoxylin–eosin staining of the testis and cauda epididymis from *Cylc1*-mutant mice and WT mice (*Figure 6—figure supplement 1*). Further examination showed that the sperm count and total motility of *Cylc1*-mutant mice were similar to those indexes in WT mice (*Figure 6E, F*). Similar to *Cylc1*-KO mice, *Cylc1*-mutant mice produced sperm with abnormal head morphologies, exhibiting varying degrees of 'fan-shaped' heads (*Figure 6G*). Compared with the closely attached acrosomes in WT mice, spermatids from *Cylc1*-mutant mice exhibited varying degrees of acrosome detachment (*Figure 6H*). The F-actin bundles of spermatid heads in WT mice possessed typical fibrous textures, whereas spermatids of *Cylc1*-mutant mice showed uneven, thin, or ectopic F-actin distribution (*Figure 6H*). The phenotype of sperm acrosome detachment in *Cylc1*-mutant mice was further observed by TEM technology (*Figure 6I*). TEM analysis of elongating spermatids within the testis revealed an enlarged space area between the IAM and the NE in *Cylc1*-mutant

**Table 2.** Semen characteristics of individuals harbouring the *CYLC1* variant.

| Individuals | Infertile for | Semen characteristics | | | | |
|---|---|---|---|---|---|---|
| | | Volume (ml) | Concentration (10⁶/ml) | Motility sperm (%) | Progressive motility (%) | Morphological normality (%) |
| AAG547 | 5 y | 1.8 | 50.2 | 49.2 | 43.1 | 3 |
| BBB654 | 3 y | 2.7 | 50.4 | 54.9 | 51 | 0 |
| BBB841 | 5 y | 4 | 52.3 | 81.1 | 62.3 | 1–2 |
| AAX199 | 3 y+ | 2.7 | 55 | 55.2 | 40.8 | 2 |
| AAX765 | 5 y | 3.2 | 4.4 | 35.9 | 19.2 | 0 |
| BBA344 | 3 y | 3 | <2 | 41.9 | 36.5 | 1 |
| BBB396 | 6 y | 1.6 | 12.8 | 30.8 | 23.1 | 0 |
| AAW228 | 5 y | 3.9 | 7.31 | 71.6 | 39.4 | 3 |
| AAT812 | 2 y | 3 | 58.7 | 21.8 | 7.6 | 3.6 |
| 3086 | 1 y | 2 | 0.9 | 30.8 | 23.1 | 0 |
| 3165 | 1 y | 1.2 | 58 | 0 | 0 | 0 |
| 3172 | 2 y | 3 | 151.5 | 14.1 | 9.6 | 1 |
| 3190 | 2 y | 2 | 22.6 | 29.5 | 26.2 | 0 |
| 3203 | 4 y | 1.2 | 24.1 | 8.9 | 6.2 | 1 |
| 3209 | 5 y | 6.4 | 10.6 | 14.2 | 4.1 | 0 |
| 3307 | 6 y | 5 | 59.9 | 77.6 | 57.1 | 4 |
| 3256 | 1 y | 1.4 | 66.6 | 67.5 | 53 | 3 |
| 3010 | 2 y | 3.4 | 54.4 | 13 | 10.2 | 1 |
| Reference values | | >1.5 | >15 | >40 | >32 | >4 |

mice (*Figure 6I*). Hence, the above animal evidence suggests that this CYLC1 variant is generally responsible for the patient phenotypes.

To explore the molecular mechanisms underlying the detached acrosomes in *Cylc1*-mutant mice, we examined the interactions between cylicin-1 variant and its partners. Although this variant did not obviously affect the protein expression of cylicin-1 and its interactors, the endogenous interaction between cylicin-1 and ACTL7A was attenuated in testis lysates of *Cylc1*-mutant mice (*Figure 6J*). The protein–protein interaction of cylicin-1 and IAM protein SPACA1 in testis lysates was also weakened in *Cylc1*-mutant mice (*Figure 6K*). Moreover, mutation of cylicin-1 affected the endogenous interaction between ACTL7A and SPACA1 (*Figure 6L*). These biochemistry experiments suggest that this variant is critical for cylicin-1 to form acrosome attachment protein complexes.

## CYLC1-associated male infertility could be rescued by intracytoplasmic sperm injection treatment

We further evaluated whether CYLC1-associated male infertility could be overcome by assisted reproductive technologies (ART). IVF experiments have indicated that the percentage of two-cell embryos using sperm of *Cylc1*-KO mice was significantly decreased compared with that of WT mice. We further conducted intracytoplasmic sperm injection (ICSI) experiments and found that two-cell embryos and blastocysts could be obtained upon ICSI using sperm from *Cylc1*-KO male mice with a similar efficiency to that from WT male mice (*Supplementary file 2*). Consistent with the observations in mice, ART for patients carrying CYLC1 variant indicated that ICSI but not IVF could serve as a promising treatment (*Table 3*). For patients AAW228, AAG547, AAX199, BBB654, BBB841, AAT812, 3307, and 3256, IVF treatments all failed, whereas ICSI treatment (with or without assisted oocyte activation AOA) generated good-quality embryos. Live birth and pregnancy was finally obtained from individuals AAW228, AAG547, AAT812, 3307, and 3256. ICSI treatments (with or without AOA) were directly performed

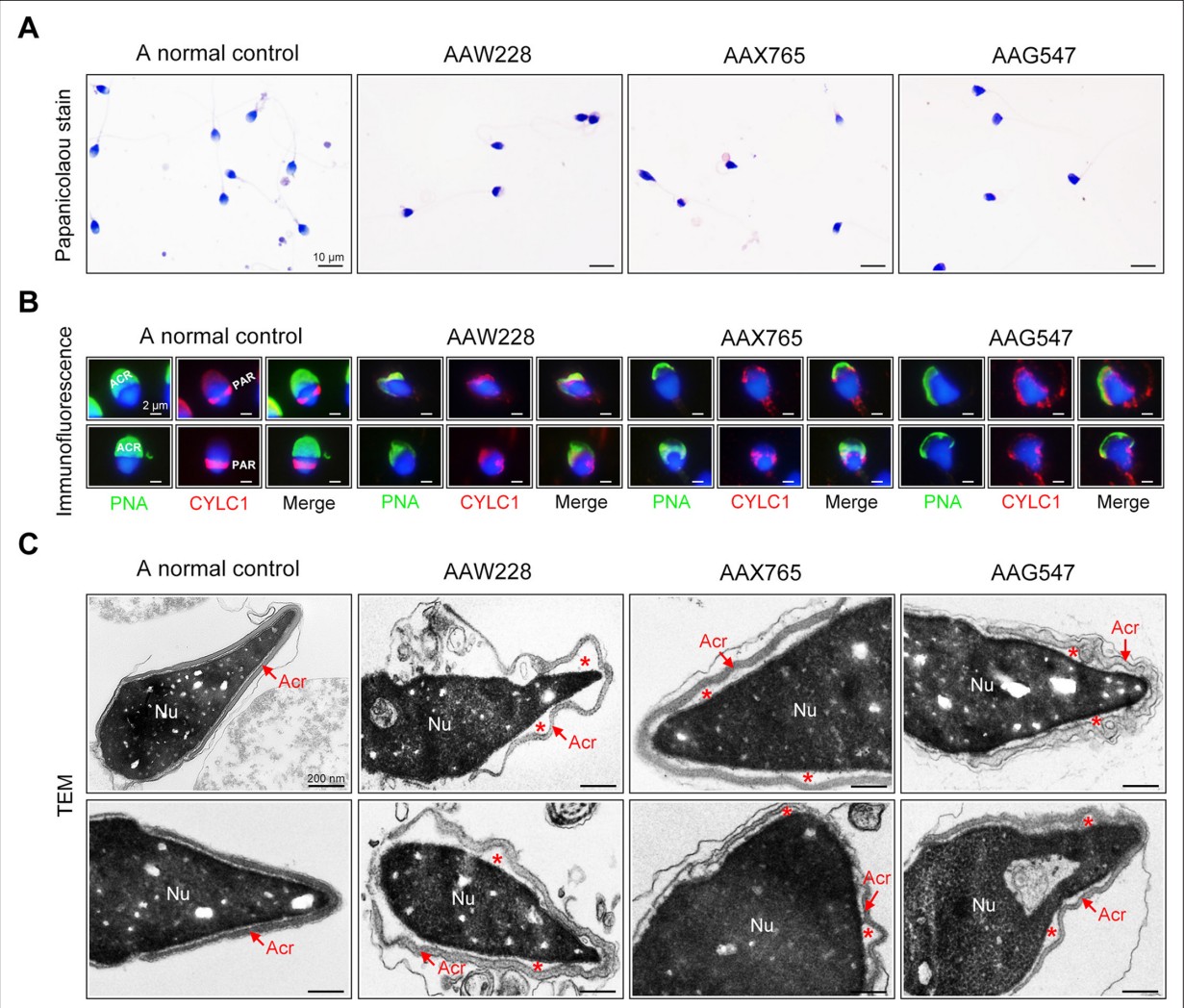

**Figure 5.** Sperm morphology and ultrastructure analyses for patients harbouring the *CYLC1* variant. (**A**) Papanicolaou staining of sperm from a fertile individual and three patients (AAW228, AAX765, and AAG547) harbouring the CYLC1 variant. Sperm from patients exhibited deformed heads. Scale bars, 10 µm. (**B**) Sperm were costained with anti-cylicin-1 antibody and peanut agglutinin (PNA)-FITC (an acrosome dye). In control sperm, Cylicin-1 was specifically localized to the postacrosomal region (PAR) and PNA-FITC-labelled acrosomes covering the anterior of heads. Both cylicin-1 distribution and acrosome structure were altered in the sperm of three patients. Scale bars, 2 µm. (**C**) Transmission electron microscopy (TEM) analysis of sperm from a normal control and three patients. Detached acrosomes of sperm were frequently observed in patient samples. The asterisk indicates the enlarged space area between the acrosome and the nucleus (Nu). Scale bars, 200 nm.

using sperm of individuals AAX765, BBA344, BBA396, 3086, 3165, 3190, 3203, 3209, and 3010, and good-quality embryos were obtained. Finally, live birth was acquired for individuals AAX765, BBA344, BBA396, 3086, 3165, 3190, 3203, and 3010 after embryo transplantation.

## Discussion

In this study, we thoroughly reveal the physiological role of cylicin-1 and explore its clinical relevance to male infertility. Cylicin-1 is specifically expressed at the PT-SAL (between the IAM and the NE) in spermatids and is ultimately distributed to the PT-PAR in sperm. *Cylc1*-KO male mice are severely subfertile with acrosome detachment and aberrant head morphology. By recruiting a large cohort (>500 infertile men with sperm head deformities), a *CYLC1* variant was identified in 19 individuals by WES. *Cylc1*-mutant mice equivalent to the variant in human *CYLC1* mimicked the phenomena of patients. Our experimental observations indicate that cylicin-1 mediates the acrosome–nucleus connection across species, and importantly, *CYLC1* variants are linked to male in/subfertility (*Figure 7*).

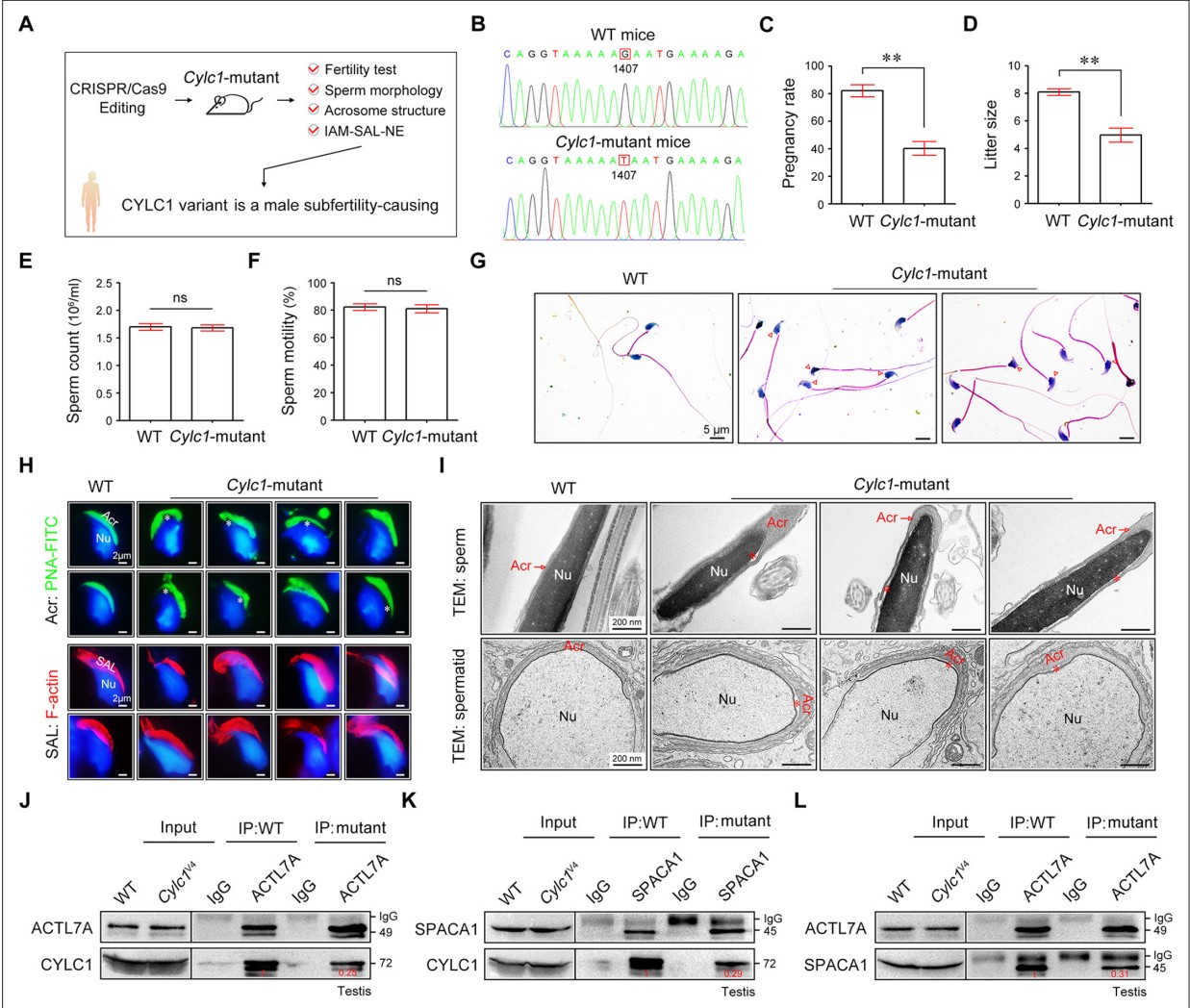

**Figure 6.** *Cylc1*-mutant mice show male subfertility with sperm acrosome detachment. (**A**) Mice carrying an SNV (c.1407G>T/p.K469N) in the *Cylc1* gene equivalent to that in nine patients harboring the CYLC1 variant were generated using CRISPR/Cas9 technology. (**B**) Sanger sequencing of the *Cylc1* gene in wild-type (WT) mice and *Cylc1*-mutant mice. (**C, D**) Adult *Cylc1*-mutant male mice and their littermate controls (*n* = 3 each group) were continuously coupled with WT female mice at a ratio of 1:2 for 2 months. (**E**) Sperm count ($10^6$/ml) of *Cylc1*-mutant mice and WT mice. (**F**) Total sperm motility (%) of *Cylc1*-mutant mice and WT mice. In C–F, data are all represented as the mean ± standard error of the mean (SEM), Student's *t* test, **p < 0.01; ns, not significant. (**G**) Sperm morphology of *Cylc1*-mutant mice and WT mice. Scale bars, 5 µm. (**H**) Visualization of the acrosome (Acr) in spermatids using the fluorescent dye peanut agglutinin (PNA)-FITC (above). Subacrosomal layer (SAL) bundles of spermatids were visible by F-actin-Tracker Red staining (below). The asterisk indicates the enlarged space area between the acrosome and the nucleus. Scale bars, 2 µm. (**I**) Transmission electron microscopy (TEM) analysis showing detachment of the acrosome from the sperm nucleus in *Cylc1*-mutant mice (above). TEM analysis of testis tissues revealing the detachment of the developing acrosome from the nucleus in spermatids of *Cylc1*-mutant mice (below). The asterisk indicates the enlarged space area between the acrosome and the nucleus. Scale bars, 200 nm. (**J**) Testis protein lysates from WT mice and *Cylc1*-mutant mice were immunoprecipitated with anti-ACTL7A antibody or rabbit IgG and then detected with anti-cylicin-1 antibody. (**K**) Testis protein lysates were immunoprecipitated with anti-SPACA1 antibody or rabbit IgG and then detected with anti-cylicin-1 antibody. (**L**) Testis protein lysates were immunoprecipitated with anti-ACTL7A antibody or rabbit IgG and then detected with anti-SPACA1 antibody.

The online version of this article includes the following source data and figure supplement(s) for figure 6:

**Source data 1.** Data used for analysis of *Figure 6C*.

**Source data 2.** Data used for analysis of *Figure 6D*.

**Source data 3.** Data used for analysis of *Figure 6E*.

**Source data 4.** Data used for analysis of *Figure 6F*.

**Source data 5.** Original file for the western blot in *Figure 6J*.

**Source data 6.** PDF containing *Figure 6J* and relevant western blot with highlighted bands and sample labels.

*Figure 6 continued on next page*

*Figure 6 continued*

**Source data 7.** Original file for the western blot in *Figure 6K*.

**Source data 8.** PDF containing *Figure 6K* and relevant western blot with highlighted bands and sample labels.

**Source data 9.** Original file for the western blot in *Figure 6L*.

**Source data 10.** PDF containing *Figure 6L* and relevant western blot with highlighted bands and sample labels.

**Figure supplement 1.** Analysis of the reproductive system of *Cylc1*-mutant mice.

**Figure supplement 1—source data 1.** Data used for analysis of *Figure 6—figure supplement 1*.

PT is a cytoskeletal element encapsulating the sperm nucleus. Compared with other sperm structures, such as acrosome and axoneme-based flagella, our understanding of the PT is very limited. Although the morphological structures and protein compositions of PT are well studied, the physiological roles of sperm PT structure are still largely unknown. Our recent two studies using *Ccin*-KO and *Actrt1*-KO mice have explored two different physiological functions of sperm PT—protecting sperm nuclear structure and mediating acrosome–nuclear connection (*Zhang et al., 2022a*; *Zhang et al., 2022b*). Loss of calicin specifically causes 'surface subsidence' of the sperm nucleus at the nuclear condensation stage, and sperm from *Ccin*-KO mice exhibit DNA damage and fertilization failure (*Zhang et al., 2022a*). ACTRT1 forms a large ARP complex with ACTL7A and ACTL9 (*Zhang et al., 2022b*). *Actrt1*-, *Actl7a*-, and *Actl9*-deficient mice all exhibit acrosome detachment and partial/total fertilization failure (*Xin et al., 2020*; *Dai et al., 2021*; *Zhang et al., 2022b*). Studies from *Actrt1*-, *Actl7a*-, *Actl9*-, and *Cylc1*-deficient mice unequivocally demonstrate the essential role of PT structure in mediating acrosome–nucleus connections. *Actrt1*-KO mice and *Cylc1*-KO mice are severely subfertile rather than completely infertile, indicating that acrosome–nucleus connections may require additive or synergistic effects of several PT proteins.

In spermatids, the acrosome is tightly bound to the PT-SAL plate, which in turn is anchored to the NE of the elongating spermatid nucleus (*Zhang et al., 2022a*). The 'IAM–SAL–NE' sandwich structure plays a central role in acrosome attachment, and perturbations in this structure are conceivably the major causes of acrosome detachment from the sperm nucleus. The existing evidence is as follows: (1) no close association of the IAM with the NE is formed in spermatids of *Spaca1*-KO mice (*Fujihara et al., 2012*); (2) SPACA1 interacts with PT-SAL proteins (e.g., CCIN, ACTRT1, and ACTL7A) to anchor the acrosome to the PA-SAL structure (*Xin et al., 2020*; *Zhang et al., 2022a*; *Zhang et al., 2022b*); and (3) PT-SAL proteins (e.g., CCIN and ACTRT1) could further connect with NE proteins (e.g., DPY19L2, FAM209, PARP11, and SPATA46) (*Zhang et al., 2022a*; *Zhang et al., 2022b*).

Cylicin-1 interacts with itself and other PT proteins (e.g., CCIN, ACTRT1, ACTRT2, ACTL7A, CAPZA3, and CAPZB) to form a PT-SAL plate. Cylicin-1 homopolymer further connects the acrosome to the NE, likely by interacting with IAM proteins (e.g., SPACA1) and NE proteins (e.g., FAM209). Disruption of the 'IAM–SAL–NE' sandwich structure is suggested to be the reason for acrosome detachment in *Cylc1*-deficient spermatids. Deficiency of cylicin-1 also partially affects the formation or stability of ACTRT1–ACTL7A and ACTL7A–SPACA1. However, the physiological organization of 'IAM–SAL–NE' protein complexes and how the CYLC1 variant will affect this structure at the molecular or even atomic level have not been clear until now.

Mutations in genes encoding PT proteins have been shown to be associated with male infertility with sperm head deformities. Xin et al. identified a homozygous missense mutation of *ACTL7A* in two infertile brothers presenting sperm acrosomal ultrastructural defects and early embryonic arrest (*Xin et al., 2020*). WES of 21 male individuals with fertilization failure identified homozygous pathogenic variants in *ACTL9* in three individuals, also showing sperm acrosome detachment (*Dai et al., 2021*). We report that three infertile men showing 'surface subsidence' of the sperm nucleus carry homozygous pathogenic mutations in *CCIN*, and the corresponding *Ccin*-mutant mice mimic the phenotype in patients (*Fan et al., 2022*).

Infertility is a very heterogeneous condition, with genetic causes involved in approximately half of cases. Taking the percentage of male in/subfertile may reach more than 20% of the population into consideration (*Gnoth et al., 2005*), the contribution of multiple genetic factors and common variants to male infertility should not be omitted. It has been suggest that mouse modelling data are critical to determine the pathogenicity of point mutations (*Singh and Schimenti, 2015*). Our mutant mouse results further proved that the identified CYLC1 variant is harmful for male fertility. Similar to *Cylc1*-KO

**Table 3.** Clinical outcomes of assisted reproductive technologies (ART) using sperm from men harbouring the CYLC1 variant.

| Individual | IVF/ICSI cycles | IVF | | ICSI | | ICSI + AOA | | Transfer cycles (embryos) | Live birth |
|---|---|---|---|---|---|---|---|---|---|
| | | Fertilization rate | Good-quality embryo rate | Fertilization rate | Good-quality embryo rate | Fertilization rate | Good-quality embryo rate | | |
| AAW228 | 2 | 0% (0/10) | - | 30% (3/10) | 0% (0/3) | 91.3% (21/23) | 66.7% (14/21) | 2 (4) | 1 |
| AAG547 | 4 | 10.5% (2/19) | 50% (1/2) | 78.9% (15/19) | 46.7 (7/15) | N/A | N/A | 3 (6) | 1 |
| AAX199 | 4 | 0% (0/2) | - | 83.3% (5/6) | 100% (5/5) | 100% (4/4) | 100% (4/4) | 5 (8) | 0 |
| BBB654 | 2 | 0% (0/14) | - | 88.9% (8/9) | 100% (8/8) | N/A | N/A | 2 (4) | 0 |
| BBB841 | 5 | 0% (0/5) | - | 50% (3/6) | 100% (3/3) | 75% (3/4) | 100% (3/3) | 2 (3) | 0 |
| AAT812 | 3 | 35.7% (5/14) | 0% (0/5) | 80% (8/10) | 50% (4/8) | 80% (4/5) | 75% (3/4) | 2 (4) | 1 |
| 3307 | 2 | 0% (1/13) | - | 100% (8/8) | 37.5% (3/8) | N/A | N/A | 1 (1) | P |
| 3256 | 1 | 0% (0/3) | - | 100% (5/5) | 60% (3/5) | N/A | N/A | 1 (1) | P |
| 3172 | 1 | 33.3% (1/3) | 0% (0/1) | N/A | N/A | N/A | N/A | 1 (1) | 0 |
| AAX765 | 7 | N/A | N/A | 72.7% (8/11) | 0% (0/8) | 100% (6/6) | 50% (3/6) | 2 (3) | 1 |
| BBA344 | 1 | N/A | N/A | N/A | N/A | 94.1% (16/17) | 37.5% (6/16) | 2 (3) | 1 |
| BBA396 | 1 | N/A | N/A | 100% (1/1) | 100% (1/1) | 75% (3/4) | 100% (3/3) | 1 (2) | 1 |
| 3086 | 1 | N/A | N/A | 100% (12/12) | 66.7% (8/12) | N/A | N/A | 2 (2) | 1 |
| 3165 | 1 | N/A | N/A | 66.7% (4/6) | 75% (3/4) | N/A | N/A | 1 (2) | 1 |
| 3190 | 1 | N/A | N/A | 93.7% (15/16) | 33.3% (5/15) | N/A | N/A | 1 (1) | 1 |
| 3203 | 1 | N/A | N/A | 83.3% (5/6) | 60% (3/5) | N/A | N/A | 1 (1) | 1 |
| 3209 | 1 | N/A | N/A | 71.4% (5/7) | 20% (1/5) | N/A | N/A | 2 (2) | 0 |
| 3010 | 1 | N/A | N/A | 70% (7/10) | 57.1% (4/7) | N/A | N/A | 1 (2) | 1 |

N/A, not available; P, pregnant.

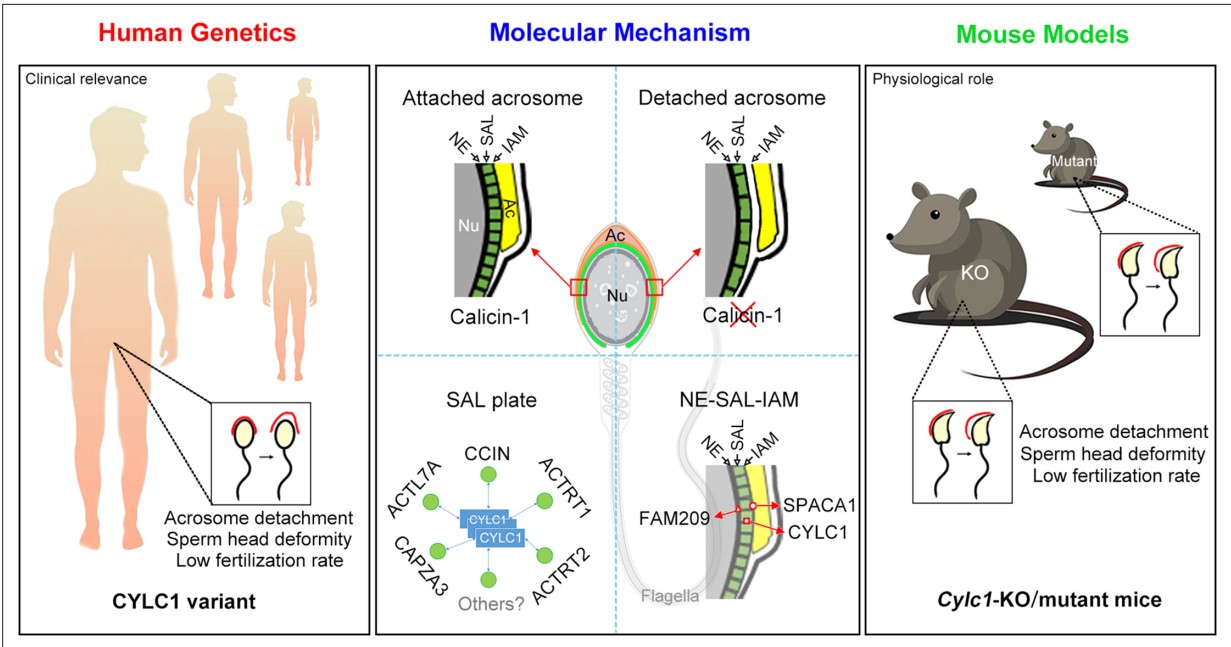

**Figure 7.** Schematic diagram showing the physiological role of CYLC1 on acrosome attachment in humans and mice. A *CYLC1* variant is identified in a large cohort of >500 Chinese infertile men with sperm head deformities. Both *Cylc1*-KO mice and *Cylc1*-mutant mice exhibited severe male subfertility with spermatozoa showing detached acrosome and head deformities. Cylicin-1 is specifically expressed at the acroplaxome (Apx), between nuclear envelope (NE) and inner acrosomal membrane (IAM), of spermatids. Cylicin-1 interacts itself and with other perinuclear theca (PT) proteins (including CCIN, ACTRT1, ACTRT2, ACTL7A, and CAPZA3) in the acroplaxome layer and mediates the acrosome–nucleus connection through interacting with IAM protein SPACA1 and NE protein FAM209.

mice, *Cylc1*-mutant mice showed severe male subfertility with sperm acrosome/head deformities. While our study was under preparation, another laboratory in parallel reported similar findings that *Cylc1*-KO mice showed sperm acrosome detachment and male subfertility as a reviewed preprint in eLife (*Schneider et al., 2023*). This report and our current study unequivocally demonstrate the essential role of cylicin-1 in mediating sperm acrosome–nucleus connections and mouse male fertility. Intriguingly, Schneider et al. indicate that deficiency of *Cylc2* (even in heterozygous state) worsened the fertility of *Cylc1*-KO mice (from severe subfertility to infertility). Extending to humans, we suggest that CYLC1 variants contribute to male infertility, especially when combined with certain variants of other male infertility-associated genes. This notion may improve our multiperspective understanding of male in/subfertility, which is a very heterogeneous condition.

After analysing the ART outcomes of the patients harbouring CYLC1 variant, we suggest that ICSI but not IVF may serve as a promising treatment for in/subfertile men carrying *CYLC1* variants. The reduced rate of acrosome reaction may be one of the underlying reasons. The outcomes of ICSI together with AOA treatment seem to be better than those of ICSI alone, because good-quality embryos are obtained in the ICSI + AOA group but not in the ICSI group for patients AAW228 and AAX765. It is still not known whether a deficiency of cylicin-1 in sperm affects oocyte activation in humans.

Taken together, we provided solid animal and clinic evidence to indicate that cylicin-1 mediate the acrosome–nucleus connection of sperm and suggest *CYLC1* variants as potential risk factors for human male fertility.

## Materials and methods
### Mice
Animal experiments were approved by the Animal Care and Use Committee of the College of Life Sciences, Beijing Normal University (CLS-AWEC-B-2023-001). WT C57BL/6J mice were obtained from Charles River Laboratory Animal Technology (Beijing, China). Exons 3 and 4 were selected as the

knockout region to generate *Cylc1*-KO mice and a G-to-T substitution at nucleotide position 1407 in *Cylc1* was created by CRISPR/Cas9 technology. In brief, single-guide RNAs (gRNAs) and/or single-stranded oligonucleotides (ssODNs) were transcribed in vitro as described previously (*Zhang et al., 2022a*). Cas9 mRNA was purchased from TriLink BioTechnologies (CA, USA). Mouse zygotes were coinjected with an RNA mixture of Cas9 mRNA, sgRNAs, and/or ssODNs. The injected zygotes were transferred into pseudopregnant recipients to obtain the F0 generation. A stable F1 generation was obtained by mating positive F0 generation mice with C57BL/6JGpt mice. The founder mice were crossed with WT C57BL/6 mice to obtain offspring. Female *Cylc1*[+/−] or *Cylc1*[+/mutant] founder mice were crossed with WT males to generate *Cylc1*-KO or *Cylc1*-mutant mice, which were tested by genotyping PCR and/or sequencing of tail genomic DNA using the Mouse Tissue Direct PCR Kit (Tiangen Biotech, Beijing, China). Primers for genotyping are provided in *Supplementary file 3*.

## Clinical cohorts

The cohort comprised more than 500 infertile patients with sperm head deformities recruited between 2015 and 2022 from the Reproductive Medicine Center, Shanghai Ninth People's Hospital of Shanghai Jiao Tong University School of Medicine (Shanghai, China) and the Center for Clinical Reproductive Medicine, The First Affiliated Hospital of Nanjing Medical University (Nanjing, China). The study regarding the cohorts was approved by the institutional review boards and signed informed consent was obtained from all subjects participating in the study (No. 20181101).

## WES and bioinformatic analysis

Genomic DNA was extracted from peripheral blood samples of patients using a QIAamp DNA Blood Mini Kit (QIAGEN, Germany). The exomes of the subject were captured by Agilent SureSelect Human All Exon V6 Enrichment kits (Agilent, CA, USA) and then sequenced on a NovaSeq platform (Illumina, CA, USA) according to the manufacturer's instructions. All reads were mapped to the human reference genome (UCSC Genome Browser hg19) using Burrows–Wheeler Alignment (*Li and Durbin, 2010*). Single-nucleotide variants and indels were detected by Genome Analysis Toolkit and then annotated by ANNOVAR (*McKenna et al., 2010*; *Wang et al., 2010*). Mutations that met the following criteria were excluded: (1) intron mutations and synonymous mutations; (2) variants with allele frequency >1% in gnomAD (total population) (*Gudmundsson et al., 2022*); (3) 'benign' or 'likely benign' variants according to the ACMG (*Gonzales et al., 2022*). Testis-specifically or testis-predominantly expressed genes were retained. The identified *CYLC1* variant in patients were verified by Sanger sequencing using the primers listed in *Supplementary file 4*.

## Expression plasmids and transient transfection

Mouse *Cylc1* cDNA was chemically synthesized by GenScript Biotech Corporation (Suzhou, China) and inserted into FLAG- or Myc-tagged pCMV vectors (Beyotime, Shanghai, China). Full-length cDNA encoding CCIN, ACTRT1, ACTL7A, SPACA1, FAM209, and SPATA46 was amplified by PCR and cloned into FLAG- or Myc-tagged pCMV vectors as described in our previous studies (*Zhang et al., 2022a*; *Zhang et al., 2022b*). Primers for CAPZA3: 5'-GGTACCATGTCAC TCAGCGTCTTGAGTAGG-3' and 5'-TCTAGATATTATCCAGTTGCACAACACAC TTC-3'.

HEK293T line was obtained from ATCC (American Type Culture Collection) and authenticated using STR profiling test by Shanghai Biowing Applied Biotechnology Co, Ltd. HEK293T cell lines were tested negative for mycoplasma contamination. HEK293T cells were cultured at 37°C in a 5% $CO_2$ incubator with Dulbecco's modified Eagle's medium (Gibco, NY, USA) with 10% fetal bovine serum (HyClone, UT, USA) and 1% penicillin–streptomycin (Gibco). Transient transfection of HEK293T cells was performed using Lipofectamine 3000 transfection reagent (Invitrogen, Shanghai, China) following the manufacturer's protocol. Cells were then harvested 48 hr after transfection.

## Generation of anti-CYLC1 antisera

The N-terminal of mouse CYLC1 was cloned into the pET-N-His-C-His vector (Beyotime, Shanghai, China) and then transfected into the ER2566 *E. coli* strain (Weidi Biotechnology, Shanghai, China). CYLC1 protein expression was induced by 1 mM IPTG (Beyotime) at 24°C for 6 hr. After centrifugation, the bacterial pellet was resuspended in buffer 1 (50 mM Tris–HCl pH 8.0, 200 mM NaCl), and the proteins were released by sonication. After centrifugation, anti-His beads (Beyotime) were added to

the supernatant and incubated for 2 hr. After washing five times with buffer 1, CYLC1-N protein was eluted with 250 mM imidazole (Beyotime). The protein was concentrated by centrifugation with ultra-filtration centrifuge tube (Millipore, Shanghai, China). One hundred micrograms of CYLC1-N protein was emulsified at a 1:1 ratio (vol/vol) with Freund's complete adjuvant (Beyotime) and administered subcutaneously into New Zealand white rabbits (Charles River) at multiple points. For the subsequent three immunizations, 50 µg CYLC1-N protein was emulsified with incomplete Freund's adjuvant (Beyotime) at an interval of 2 weeks. One week after the last immunization, blood was collected, and the serum was separated.

## Mouse fertility testing

To confirm the fertility of male mice, natural mating tests were conducted. Briefly, three *Cylc1*-KO (or *Cylc1*-mutant) and three littermate control sexually mature male mice (8–12 weeks old) were paired with two 8- to 10-week-old C57BL/6J females (each male was mated with two female mice) for more than 2 months. The vaginal plugs of the mice were examined every morning. Then, the female mice with vaginal plugs were separately fed, and the number of pups per litter was recorded.

## Semen characteristics analysis

Semen samples of human subjects were collected through masturbation after 2–7 days of sexual abstinence and analysed in the source laboratories as part of the routine biological examination according to the 5th World Health Organization (WHO) guidelines. For mouse models, the backflushing method was used to retrieve sperm cells from the cauda epididymis (*Baker et al., 2014*). Sperm counts were determined using a fertility counting chamber (Makler, Haifa, Israel) under a light microscope, and sperm mobility was assessed via the application of a computer-assisted sperm analysis system. Sperm were collected from the cauda epididymis and washed three times in phosphate-buffered saline (PBS) buffer. The sperm suspension was mounted on a glass slide, air-dried, and fixed with 4% paraformaldehyde (PFA) for 10 min at room temperature. The slides were stained with Papanicolaou solution (Solarbio, Beijing, China) and observed using a DM500 optical microscope (Leica, Germany).

## Cell staining

Spermatogenic cells were isolated from adult mouse testes by digestion with 1 mg/ml collagenase IV, 1 mg/ml hyaluronidase, 1 mg/ml trypsin, and 0.5 mg/ml DNase I (Solarbio) for 20 min on a shaker. A cell suspension containing spermatids was mounted on a glass slide, air-dried, and fixed with 4% PFA for 10 min at room temperature. After permeabilization with 1% Triton X-100 for 30 min, the slides of sperm and spermatids were blocked with 5% goat serum for 45 min. Anti-cylicin-1 antibody was added to the slide and incubated overnight at 4°C. After washing three times with PBS, slides were incubated with Alexa Fluor 484-labelled donkey anti-rabbit IgG for 1 hr at room temperature. For the staining of the acrosome, PT-SAL and manchette, PNA-FITC dye, Actin-Tracker Red-555 dye, and Tubulin-Tracker Red dye were used. The nuclei were counterstained with DAPI dye.

## Transmission electron microscope

Precipitation of mouse sperm and testis tissues ($\sim 1$ mm$^3$) were fixed with 2.5% (vol/vol) glutaraldehyde in 0.1 M phosphate buffer (PB) (pH 7.4) for 24 hr at 4°C. As a previous protocol (*Zhang et al., 2022a*), the samples were washed four times in PB and first immersed in 1% (wt/vol) OsO4 and 1.5% (wt/vol) potassium ferricyanide aqueous solution at 4°C for 2 hr. After washing, the samples were dehydrated through graded alcohol into pure acetone. Samples were infiltrated in a graded mixture (3:1, 1:1, 1:3) of acetone and SPI-PON812 resin (21 ml SPOPON812, 13 ml DDSA (dodecyl succinic anhydride) and 11 ml NMA (N-methylolacrylamide)), and then the pure resin was changed. The specimens were embedded in pure resin with 1.5% BDMA (benzyl dimethylamine), polymerized for 12 hr at 45°C and 48 hr at 60°C, cut into ultrathin sections (70 nm thick), and then stained with uranyl acetate and lead citrate for subsequent observation and photography with a Tecnai G2 Spirit 120 kV (FEI) electron microscope. All reagents were purchased from Zhongjingkeyi Technology (Beijing, China).

## Scanning electron microscope

Mouse sperm from the cauda epididymis were fixed in 2.5% phosphate-buffered glutaraldehyde (GA) (Zhongjingkeyi Technology, Beijing, China) at room temperature for 30 min and then deposited on

coverslips. As described in a previous protocol (*Zhang et al., 2022a*), the coverslips were dehydrated via an ascending gradient of 50%, 70%, 95%, and 100% ethanol and air-dried. Specimens were then attached to specimen holders and coated with gold particles using an ion sputter coater before being viewed with a JSM-IT300 scanning electron microscope (JEOL, Tokyo, Japan).

## In vitro fertilization

As described in a previous protocol (*Zhang et al., 2022a*), 6-week-old C57BL/6J female mice were superovulated by injecting 5 IU (0.1 ml) of pregnant mare serum gonadotropin (PMSG), followed by 5 IU (0.1 ml) of human chorionic gonadotropin (hCG) 48 hr later. The sperm were released from the cauda epididymis of 10-week-old male mice, and sperm capacitation was performed for 50 min using TYH solution. Cumulus–oocyte complexes (COCs) were obtained from the ampulla of the uterine tube at 14 hr after hCG injection. The ampulla was torn with a syringe needle, and the COCs were gently squeezed onto liquid drops of HTF (human tubal fluid) medium. COCs were then incubated with 5–10 µl sperm suspension (sperm concentration: $1–5 \times 10^6$) in HTF liquid drops at 37 °C under 5% $CO_2$. After 6 hr, the eggs were washed several times using HTF medium to remove the cumulus cells and then transferred to liquid drops of KSOM (K+ simplex optimised medium) medium. Two-cell embryos were counted at 24 hr postfertilization. All lipid drops were covered with mineral oil and equilibrated overnight at 37°C under 5% $CO_2$. All reagents were purchased from Aibei Biotechnology (Nanjing, China).

## Intracytoplasmic sperm injection

One-month-old C57BL/6J female mice were superovulated by administration of 10 IU PMSG combined with 10 IU hCG (48 hr later). Oocytes were obtained from the ampulla of the uterine tube at 14 hr after hCG injection. Mouse sperm heads were separated from sperm tails and injected into mouse oocytes using a Piezo-driven pipette (PrimeTech, Osaka, Japan). The injected oocytes were cultured in KSOM medium at 37°C under 5% $CO_2$. All reagents were purchased from Nanjing Aibei Biotechnology.

## Acrosome reaction

Sperm were collected from the cauda epididymis and capacitated for 50 min in TYH medium at 37°C under 5% $CO_2$. Highly motile sperm were collected from the upper portion of the medium, and 10 µM calcium ionophore A23187 (Sigma, Germany) was added to induce the acrosome reaction. After 15 min, sperm were spotted on a glass microscope slide, dried, and fixed with 4% PFA for 10 min. Acrosomes were stained with PNA-FITC dye, and nuclei were labelled with DAPI.

## Coimmunoprecipitation

As described in a previous protocol (*Zhang et al., 2022a*), 48 hr after transfection, HEK293T cells were lysed with Pierce IP Lysis Buffer (Thermo Fisher, 87787) with protease inhibitor cocktail for 30 min at 4°C and then centrifuged at $12,000 \times g$ for 10 min. Pierce IP Lysis Buffer was composed of 25 mM Tris–HCl pH 7.4, 150 mM NaCl, 1 mM EDTA (ethylene diamine tetraacetic acid), 1% NP-40 and 5% glycerol. To prepare input samples, protein lysates were collected and boiled for 5 min in 1.2× sodium dodecyl sulphate (SDS) loading buffer. The lysates were precleared with 10 µl Pierce Protein A/G-conjugated Agarose (Thermo Fisher) for 1 hr at 4°C. Precleared lysates were incubated overnight with 2 µg anti-Myc antibody or anti-FLAG antibody at 4°C. The lysates were then incubated with 20 µl Pierce Protein A/G-conjugated Agarose for 2 hr at 4°C. The agarose beads were washed four times with Pierce IP Lysis Buffer and boiled for 5 min in SDS loading buffer. Input and IP samples were analysed by western blotting using anti-FLAG or anti-Myc antibodies. For endogenous co-IP, adult mouse testis tissues were lysed with Pierce IP Lysis Buffer. Precleared lysates were separated into two groups: one group was treated with 2 µg anti-ACTL7A antibody, anti-FAM209 antibody, or anti-SPACA1 antibody, and another group (negative control) was treated with 2 µg rabbit IgG. Other endogenous co-IP procedures were similar to the co-IP assay in HEK293T cells. Antibodies used in this study are listed in *Supplementary file 5*.

## Microscale thermophoresis

The procedure of prokaryotic expression and purification of CYLC1 is described in the paragraph 'generation of anti-CYLC1 antisera'. MST experiments were conducted on a Monolith NT.115 system

(NanoTemper Technologies, Germany). GFP-labelled ACTL7A, SPACA1, or FAM209 was transfected into HEK293T cells and whole proteins were extracted and then added to 16 PCR tubes (10 µl per tube). The purified ligand CYLC1 was diluted using MST buffer (50 mM Tris–HCl pH 7.5, 150 mM NaCl, 10 mM $MgCl_2$, 0.05% (vol/vol) Tween 20) into 16 gradients. Ten microliters of different concentrations of ligand CYLC1 were mixed with 10 µl of $^{GFP-}$ACTL7A, $^{GFP-}$SPACA1, and $^{GFP-}$FAM209 cell lysates and reacted in a dark box at room temperature for 30 min. The samples were added to monolith capillaries (NanoTemper) and subsequently subjected to MST analysis.

## Western blotting

As described in a previous protocol (*Zhang et al., 2022a*), proteins from HEK293T cells were extracted using RIPA (radioimmunoprecipitation assay) lysis buffer (Applygen, Beijing, China) containing 1 mM phenylmethylsulfonyl fluoride and protease inhibitors on ice. For sperm samples, 1% SDS was added to RIPA lysis buffer. The supernatants were collected following centrifugation at 12,000 × *g* for 20 min. Proteins were electrophoresed in 10% SDS–polyacrylamide gel electrophoresis gels and transferred to nitrocellulose membranes (GE Healthcare). The blots were blocked in 5% milk and incubated with primary antibodies overnight at 4°C, followed by incubation with anti-rabbit or anti-mouse IgG H&L (HRP (horseradish peroxidase)) (Abmart) at a 1/10,000 dilution for 1 hr. The signals were evaluated using Super ECL Plus Western Blotting Substrate (Applygen) and a Tanon5200 Multi chemiluminescence imaging system (Tanon, Shanghai, China). Antibodies used in this study are listed in *Supplementary file 5*.

## Statistical analysis

The data are presented as the mean ± standard error of the mean and compared for statistical significance using GraphPad Prism version 5.01 (GraphPad Software). Unpaired, two-tailed Student's *t* test was used for the statistical analyses. Statistically significant differences are represented as *$p < 0.05$, **$p < 0.01$, and ***$p < 0.001$.

## Acknowledgements

We acknowledge Xi-Xia Li and Shuang-Zhong Lv from the Center for Biological Imaging (CBI), Institute of Biophysics, Chinese Academy of Sciences for their help in making TEM samples.

---

## Additional information

### Funding

| Funder | Grant reference number | Author |
| --- | --- | --- |
| National Natural Science Foundation of China | 32370905 | Su-Ren Chen |
| National Natural Science Foundation of China | 81901531 | Yong Fan |
| National Natural Science Foundation of China | 81971374 | Xiaoyu Yang |
| National Natural Science Foundation of China | 81871163 | Qifeng Lyu |
| National Natural Science Foundation of China | 82171685 | Ling Wu |
| National Natural Science Foundation of China | 82271732 | Yanping Kuang |
| National Natural Science Foundation of China | 81971448 | Bin Li |
| National Key Research and Development Program of China | 2019YFA0802101 | Su-Ren Chen |

| Funder | Grant reference number | Author |
|---|---|---|
| National Key Research and Development Program of China | 2022YFC2702702 | Mingxi Liu |
| Open Fund of Key Laboratory of Cell Proliferation and Regulation Biology, Ministry of Education | | Su-Ren Chen |
| Fund for Excellent Young Scholars of Shanghai Ninth People's Hospital, Shanghai Jiao Tong University School of Medicine | JYYQ004 | Yong Fan |
| Natural Science Foundation of Jiangsu Province | BK20230004 | Mingxi Liu |

The funders had no role in study design, data collection, and interpretation, or the decision to submit the work for publication.

## Author contributions

Hui-Juan Jin, Yong Fan, Xiaoyu Yang, Investigation, Methodology; Yue Dong, Xiao-Zhen Zhang, Resources, Methodology; Xin-Yan Geng, Software; Zheng Yan, Ling Wu, Meng Ma, Bin Li, Qifeng Lyu, Yun Pan, Resources; Mingxi Liu, Yanping Kuang, Supervision, Funding acquisition; Su-Ren Chen, Supervision, Funding acquisition, Project administration, Writing – review and editing

## Author ORCIDs

Yong Fan https://orcid.org/0000-0001-9290-184X
Mingxi Liu https://orcid.org/0000-0001-6499-7899
Su-Ren Chen https://orcid.org/0000-0002-9337-5412

## Ethics

The study regarding the cohorts was approved by the institutional review boards and signed informed consent was obtained from all subjects participating in the study (No. 20181101).
Animal experiments were approved by the Animal Care and Use Committee of the College of Life Sciences, Beijing Normal University (CLS-AWEC-B-2023-001).

Reviewer #1 (Public Review): https://doi.org/10.7554/eLife.95054.2.sa1
Reviewer #2 (Public Review): https://doi.org/10.7554/eLife.95054.2.sa2
Author response https://doi.org/10.7554/eLife.95054.2.sa3

# Additional files

## Supplementary files

• Supplementary file 1. WES reveals variants in known sperm head deformity-associated genes in infertile men.

• Supplementary file 2. Intracytoplasmic sperm injection (ICSI) experiments using sperm from wild-type (WT) mice or *Cylc1*-KO mice.

• Supplementary file 3. Primers for mouse genotyping.

• Supplementary file 4. Primers for Sanger sequencing of *CYLC1* in humans.

• Supplementary file 5. Antibodies used in this study.

• MDAR checklist

## Data availability

All data generated or analysed during this study are included in the manuscript and supporting files; source data files have been provided for *Figures 1–3 and 6* as well as *Figure 1—figure supplements 2 and 4*, *Figure 3—figure supplements 1 and 2*, and *Figure 6—figure supplement 1*.

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
