## [Editor Report · eLife assessment]

Spermiogenesis is a complex process allowing the emergence of specific sperm organelle, including the acrosome, a sperm giant vesicle of secretion. This **important** study reports the key role of Cylicin-1 in acrosome biogenesis and identifies the molecular partners necessary for acrosome anchoring. The **compelling** demonstration is based on infertile patient samples and two animal models. Overall, this provides results that will be invaluable to the male reproduction community, including scientists and andrologists.

---

## [Referee Report · Reviewer #1 (Public Review)]

Summary:

The study investigates the role of cylicin-1 (CYLC1) in sperm acrosome-nucleus connections and its clinical relevance to male infertility. Using mouse models, the researchers demonstrate that cylicin-1 is specifically expressed in the post acrosomal sheath-like region in spermatids and plays a crucial role in mediating acrosome-nucleus connections. Loss of CYLC1 results in severe male subfertility, characterized by acrosome detachment and aberrant head morphology in sperm. Further analysis of a large cohort of infertile men reveals CYLC1 variants in patients with sperm head deformities. The study provides valuable insights into the role of CYLC1 in male fertility and proposes CYLC1 variants as potential risk factors for human male infertility, emphasizing the importance of mouse models in understanding the pathogenicity of such variants.

Strengths:

This article demonstrates notable strengths in various aspects. Firstly, the clarity and excellent writing style contribute to the accessibility of the content. Secondly, the employed techniques are not only relevant but also complementary, enhancing the robustness of the study. The precision in their experimental design and the meticulous interpretation of results reflect the scientific rigor maintained throughout the study. Furthermore, the decision to create a second mouse model with the exact CYLC1 mutation found in humans adds significant qualitative value to the research. This approach not only validates the clinical relevance of the identified variant but also strengthens the translational impact of the findings.

Weaknesses:

There are no obvious weaknesses. While a few minor refinements, as suggested in the recommendations to authors, could enhance the overall support for the data and the authors' messages, these suggested improvements in no way diminish the robustness of the already presented data.

---

## [Referee Report · Reviewer #2 (Public Review)]

Summary:

* To verify the function of PT-associated protein CYLC1, the authors generated a Cylc1-KO mouse model and revealed that loss of cylicin-1 leads to severe male subfertility as a result of sperm head deformities and acrosome detachment.

* Then they also identified a CYLC1 variant by WES analysis from 19 infertile males with sperm head deformities.

* To prove the pathogenicity of the identified mutation site, they further generated Cylc1-mutant mice that carried a single amino acid change equivalent to the variant in human CYLC1. The Cylc1-mutant mice also exhibited male subfertility with detached acrosomes of sperm cells.

Strengths:

* The phenotypes observed in the Cylc1-KO mice provide strong evidence for the function of CYLC1 as a PT-associated protein in spermatogenesis and male infertility.

* Further mechanistic studies indicate that loss of cylicin-1 in mice may disrupt the connections between the inner acrosomal membrane and acroplaxome, leading to detached acrosomes of sperm cells.

Weaknesses:

* The authors identified a missense mutation (c.1377G>T/p. K459N) from 19 infertile males with sperm head deformities. The information for the variant in Table 1 is insufficient to determine the pathogenicity and reliability of the mutation site. More information should be added, including all individuals in gnomAD, East Asians in gnomAD, 1000 Genomes Project for allele frequency in the human population; MutationTaster, M-CAP, FATHMM, and more other tools for function prediction. Then, the expression of CYLC1 in the spermatozoa from men with CYLC1 mutation should be explored by qPCR, Western blot, or IF staining analyses.

* Although 19 infertile males were found carrying the same missense mutation (c.1377G>T/p. K459N), their phenotypes are somewhat different. For example, sperm concentrations for individuals AAX765, BBA344, and 3086 are extremely low but this is not observed in other infertile males. Then, progressive motility for individuals AAT812, 3165, 3172, 3203, and 3209 are extremely low but this is also not observed in other infertile males. It is worth considering why different phenotypes are observed in probands carrying the same mutation.

---

## [Author Response]

**Reviewer #1 (Public Review):**
Summary:The study investigates the role of cylicin-1 (CYLC1) in sperm acrosome-nucleus connections and its clinical relevance to male infertility. Using mouse models, the researchers demonstrate that cylicin-1 is specifically expressed in the post acrosomal sheath-like region in spermatids and plays a crucial role in mediating acrosome-nucleus connections. Loss of CYLC1 results in severe male subfertility, characterized by acrosome detachment and aberrant head morphology in sperm. Further analysis of a large cohort of infertile men reveals CYLC1 variants in patients with sperm head deformities. The study provides valuable insights into the role of CYLC1 in male fertility and proposes CYLC1 variants as potential risk factors for human male infertility, emphasizing the importance of mouse models in understanding the pathogenicity of such variants.

We appreciate the comprehensive summary of reviewer 1.

Strengths:This article demonstrates notable strengths in various aspects. Firstly, the clarity and excellent writing style contribute to the accessibility of the content. Secondly, the employed techniques are not only relevant but also complementary, enhancing the robustness of the study. The precision in their experimental design and the meticulous interpretation of results reflect the scientific rigor maintained throughout the study. Furthermore, the decision to create a second mouse model with the exact CYLC1 mutation found in humans adds significant qualitative value to the research. This approach not only validates the clinical relevance of the identified variant but also strengthens the translational impact of the findings.

We appreciate the positive comment of reviewer 1.

Weaknesses:There are no obvious weaknesses. While a few minor refinements, as suggested in the recommendations to authors, could enhance the overall support for the data and the authors' messages, these suggested improvements in no way diminish the robustness of the already presented data.

In the recommendation for the authors, reviewer 1 mentioned a recent study (Schneider et al., eLife, 2023) showing that Cylc1-KO mice exhibits a reduced sperm count, an observation not noted in our current study. We would like to comment that that main and most important phenotype of Cylc1-KO mice in both studies is quite similar, including male subfertility and abnormal head morphology. We think the different targeting strategy and mouse strain may cause this discrepancy. In Schneider’s and our current studies, the total motility abnormality of Cylc1-KO mice are not observed. We appreciate the suggestion of reviewer 1 to further examine the detailed parameters of motility such as VCL, VSL, and ALH. Given that the head deformation is the most obvious phenotype of Cylc1-KO mice and the focus of our study, we feel sorry that this detailed analysis of sperm motility was not performed in the current stage. Reviewer 1 also asked whether Cylc1-KO female mice are fertile or not. Given that Cylc1 is an X chromosome gene and Cylc1-KO (Cylc1-/Y) mice are severely subfertile, we do not obtain enough Cylc1-KO female mice to examine their fecundity. We also would like to thank reviewer 1 to point out several inaccurate descriptions.

**Reviewer #2 (Public Review):**
Summary:To verify the function of PT-associated protein CYLC1, the authors generated a Cylc1-KO mouse model and revealed that loss of cylicin-1 leads to severe male subfertility as a result of sperm head deformities and acrosome detachment. Then they also identified a CYLC1 variant by WES analysis from 19 infertile males with sperm head deformities. To prove the pathogenicity of the identified mutation site, they further generated Cylc1-mutant mice that carried a single amino acid change equivalent to the variant in human CYLC1. The Cylc1-mutant mice also exhibited male subfertility with detached acrosomes of sperm cells.

We appreciate the comprehensive summary of reviewer 2.

Strengths:The phenotypes observed in the Cylc1-KO mice provide strong evidence for the function of CYLC1 as a PT-associated protein in spermatogenesis and male infertility. Further mechanistic studies indicate that loss of cylicin-1 in mice may disrupt the connections between the inner acrosomal membrane and acroplaxome, leading to detached acrosomes of sperm cells.

We appreciate the positive comment of reviewer 2.

Weaknesses:The authors identified a missense mutation (c.1377G>T/p. K459N) from 19 infertile males with sperm head deformities. The information for the variant in Table 1 is insufficient to determine the pathogenicity and reliability of the mutation site. More information should be added, including all individuals in gnomAD, East Asians in gnomAD, 1000 Genomes Project for allele frequency in the human population; MutationTaster, M-CAP, FATHMM, and more other tools for function prediction. Then, the expression of CYLC1 in the spermatozoa from men with CYLC1 mutation should be explored by qPCR, Western blot, or IF staining analyses. Although 19 infertile males were found carrying the same missense mutation (c.1377G>T/p. K459N), their phenotypes are somewhat different. For example, sperm concentrations for individuals AAX765, BBA344, and 3086 are extremely low but this is not observed in other infertile males. Then, progressive motility for individuals AAT812, 3165, 3172, 3203, and 3209 are extremely low but this is also not observed in other infertile males. It is worth considering why different phenotypes are observed in probands carrying the same mutation.

We appreciate the suggestion of reviewer 2. First, Table 1 shows the information of the variant identified in CYLC1 gene, including allele frequency in gnomAD and functional prediction by SIFT, PolyPhen-2, and CADD. Given that mutant mice is a gold standard to confirm the pathogenicity of a variant, we generate Cylc1-mutant mice and Cylc1-mutant mice exhibit male subfertility with sperm acrosome detachment. The animal evidence is much more solid than bioinformatics prediction to confirm the pathogenicity of the identified variant in the CYLC1 gene. Second, the expression of CYLC1 in the spermatozoa from patients have been examined by IF staining (Fig. 5B). Unfortunately, the patients declined to continue in the project to donate more semen for qPCR and Western blot analyses. Third, the reviewer 2 asks why not all patients with CYLC1 gene mutation show the identical phenotype. Although some patients exhibit low sperm count or reduced motility, sperm head deformities are the shared phenotype of 19 patients. Many factors, such as way of life, may affect sperm quality. Perfectly identical phenotype of all 19 patients carrying the CYLC1 mutation is idealistic and will not always happen in clinical diagnosis. We also appreciate other suggestions from reviewer 2.